# Opioid-free anesthesia compared to opioid anesthesia for lung cancer patients undergoing video-assisted thoracoscopic surgery: A randomized controlled study

Guangquan An[1][⊙], Yiwen Zhang[2][⊙], Nuoya Chen[2¤a], Jianfeng Fu[2], Bingsha Zhao[2¤b], Xuelian Zhao[ID][2]*

**1** Department of Second Surgery, The Fourth Hospital of Hebei Medical University, Shijiazhuang, Hebei, PR China, **2** Department of Anesthesia, The Fourth Hospital of Hebei Medical University, Shijiazhuang, Hebei, PR China

⊙ These authors contributed equally to this work.
¤a Current address: Department of Anesthesia, Hengshui People's Hospital, Hebei, PR China
¤b Current address: Department of Anesthesia, Tianjin Chest Hospital, Tianjin, PR China
* zhaoxuelian@hebmu.edu.cn

**Data Availability Statement:** All relevant data are within the paper.

## Abstract

### Background

Reducing intra-operative opioid consumption benefits patients by decreasing postoperative opioid-related adverse events. We assessed whether opioid-free anesthesia would provide effective analgesia-antinociception monitored by analgesia index in video-assisted thoracoscopic surgery.

### Methods

Patients (ASA I-II, 18–65 years old, BMI <30 kg m$^{-2}$) scheduled to undergo video-assisted thoracoscopic surgery under general anesthesia were randomly allocated into two groups to receive opioid-free anesthesia (group OFA) with dexmedetomidine, sevoflurane plus thoracic paravertebral blockade or opioid-based anesthesia (group OA) with remifentanil, sevoflurane, and thoracic paravertebral blockade. The primary outcome variable was pain intensity during the operation, assessed by the depth of analgesia using the pain threshold index with the multifunction combination monitor HXD-I. Secondary outcomes included depth of sedation monitoring by wavelet index and blood glucose concentration achieved from blood gas.

### Results

One hundred patients were randomized; 3 patients were excluded due to discontinued intervention and 97 included in the final analysis. Intraoperative pain threshold index readings were not significantly different between group OFA and group OA from arriving operation room to extubation (P = 0.86), while the brain wavelet index readings in group OFA were notably lower than those in group OA from before general anesthesia induction to recovery

**Funding:** The authors received no specific funding for this work.

**Competing interests:** The authors have declared that no competing interests exist.

of double lungs ventilation ($P$ <0.001). After beginning of operation, the blood glucose levels in group OFA increased compared with baseline blood glucose values ($P$ < 0.001). The recovery time and extubation time in group OFA were significantly longer than those in group OA ($P$ <0.007).

## Conclusions

This study suggested that our OFA regimen achieved equally effective intraoperative pain threshold index compared to OA in video-assisted thoracoscopic surgery. Depth of sedation was significantly deeper and blood glucose levels were higher with OFA. Study's limitations and strict inclusion criteria may limit the external validity of the study, suggesting the need of further randomized trials on the topic.

**Trial registration:** ChiCTR1800019479, Title: "Opioid-free anesthesia in video-assisted thoracoscopic surgery lobectomy".

## Introduction

Pain management following video-assisted thoracoscopic surgery (VATS) is still a challenging issue associated with significant pain and a risk of developing chronic pain due to intercostal nerve compression and injuries that occur when instrumentation is used in the intercostal space [1]. Opioids are the most effective analgesics and are widely used in VATS with opioid based anesthesia (OA) to produce perioperative analgesia, hypnosis, and the inhibition of the sympathetic system. Side effects associated with intraoperative opioid administration include hyperanalgesia [2], chronic postoperative pain, respiratory depression [3], postoperative nausea and vomiting (PONV) [4] and postoperative delirium [5]. The possibility of higher risks of progression and metastasis of lung cancer cells treated with opioids has also been hypothesized [6–9]. This explains the current trend of using non-opioid drugs as an alternative to the use of opioids for pain management during the perioperative period.

Recent studies have shown that opioid-free anesthesia (OFA) with a multimode intraoperative opioid sparing strategy, such as locoregional analgesia, ketamine, clonidine, and dexmedetomidine, can provide better postoperative analgesia, as well as lower risks of PONV and hypoxemia in laparoscopic cholecystectomy, breast cancer surgery, spine surgery, and bariatric surgery [10–13]. In a retrospective study, Bello and colleagues reported that OFA for patients undergoing thoracotomy appeared to be feasible and enhance postoperative pain relief with less cumulative ropivacaine consumption in the patient-controlled epidtlral analgesia machine for 48 h after surgery, in addition to less postoperative morphine consumption and lower postoperative VAS scores [14]. In another retrospective study, Guinot and colleagues showed that OFA in cardiac surgery was associated with lower morphine consumption, shorter intubation times and shorter ICU stays [15]. We hypothesized that the use of OFA in patients receiving VATS can achieve the goals of analgesia, hypnosis, hemodynamic stability, and the avoidance of the side effects of opioids. Few studies have reported on the pain management related to VATS with OFA and the continuous monitoring of the depth of analgesia during VATS.

A variety of analgesia monitors have been developed to determine the depth of pain during surgery [16]. The pain threshold index (PTI), which is a 0–100 noninvasive index calculated from changes in electroencephalographic (EEG) signals, has also been recently proposed to reflect the analgesia-nociception balance during general anesthesia (GA). This equipment can

offer a continuous analgesia-nociception index reading in unconscious patients under GA [17, 18]. There are currently few trials in the literature that have investigated whether OFA provides a uniform balance of analgesia-nociception when compared to traditional opioid-anesthesia in VATS with PTI.

In this prospective randomized trial, we tested the main hypothesis that protocol-driven intraoperative OFA can provide effective intraoperative analgesia compared to opioid-based GA with PTI.

## Materials and methods

### Trial design

This prospective, randomized, parallel-group and single-center clinical trial was conducted from November 2018 to March 2019, at Fourth Hospital of Hebei Medical University China. The study complied with the Declaration of Helsinki and was approved by the local ethics committee (Fourth Hospital of Hebei Medical University Ethics Committee, Hebei, China, #2018049, Chairperson Prof Guiying Wang, on 20 Nov 2018). The study was registered on ClinicalTrials.gov (ChiCTR 1800019479).

### Participant eligibility and consent

We included patients with ASA physical status I or II, aged 18–65 years who were scheduled for elective thoracoscopic radical resection of lung cancer. Exclusion criteria were: pregnancy; breastfeeding; allergy to any experimental drug or its excipients; β-blockers therapy and HR <50 bpm; body mass index (BMI) more than 30 kg m$^{-2}$; central nervous system diseases (such as epilepsy, cerebral infarction, or cerebral hemorrhage history); and history of chronic pain, alcohol, or drug abuse. The eligible patient signed a written informed consent form after obtaining the consent of the participant.

### Allocation and blinding

Patients were randomized in two groups (group OA and group OFA) using computer-generated random numbers (Excel® version 16). The distribution results are sealed in opaque, sealed envelopes and kept by the research coordinator. On the day of operation, the research coordinator gave each patient an envelope to the anesthesiologist in the operation room. To facilitate the management of intraoperative anesthesia, anesthesiologists who did not participate in patient evaluation at any time are aware of this plan. In the operation room, the data collector who records various outcomes during general anesthesia knows the method of anesthesia. Patients, surgical staff, and postoperative evaluators were unaware of their groupings.

### EEG measurement method

After the patient rested for 2 min, the forehead and the mastoid behind the ears were cleaned. The EEG electrodes of the multifunction combination monitor HXD-I (Heilongjiang Huaxiang Technology Co., Ltd., Heilongjiang, China) were placed on 2 cm above the midpoint between the eyebrows, above the bilateral eyebrows, and at the bilateral mastoid process. Two channels of EEG data were recorded by HXD-I and were then implemented to reduce the complexity of the features through continuous and discrete wavelet transform. The repeatable and regular changes when pain present were extracted from the brain waves as the characteristic indicator of subjective pain.

The analgesic index PTI calculated according to changes in EEG signals, specifically reflects the tolerance of the cerebral cortex to painful stimuli. It is suitable for GA but not suitable for

local anesthesia. The range is 0–100. A PTI < 39 may be a risk of overdose of analgesics; A PTI 40–60 is appropriate for intraoperative analgesia; A PTI >70 denotes insufficient analgesia and need to add analgesics. The PTI increases following painful stimuli and decreases with administration of analgesics.

The sedation index (wavelet index, WLI) ranging from 0–100 shows the depth of sedation with changes in EEG signals. WLI < 35 over sedation; WLI 35–69 narcotization; WLI 70–89 light narcosis/ sleep; WLI > 90 wake fulness.

## Anesthetic management protocol (Table 1)

On arrival at the operating room, the overnight fasting patients received standard monitoring, including ECG, invasive blood pressure, pulse oximetry, temperature. The intraoperative analgesic administration regimen was variable between the groups (Table 1):

- For patients in the group OFA, a loading dose of dexmedetomidine (1 µg kg$^{-1}$ over 10 min) and a bolus of atropine 0.5 mg were administered before GA induction. GA was maintained with dexmedetomidine infusion (0.5 µg kg$^{-1}$h$^{-1}$) terminated 40 min before the end of surgery. All patients received ketorolac 30 mg IV at GA induction.

- For patients in the group OA, a bolus of sufentanil (0.5 µg kg$^{-1}$) IV was administered for GA induction. For maintenance of GA, remifentanil (25 µg ml$^{-1}$, 8–20 ml h$^{-1}$) was continuously infused and discontinued before skin closure.

In both groups, GA was induced with a bolus of etomidate (0.2–0.3mg kg$^{-1}$), followed by cisatracurium 0.2mg kg$^{-1}$ for muscle relaxation. After that, a single-lumen endotracheal tube (ETT) and a bronchial blocker (BB) were inserted into airway in turn. Muscle relaxation was maintained with intermittent intravenous injection boluses of cisatracurium (2–4 mg per 30 min). Sevoflurane was adjusted to maintain anesthesia. The use of midazolam was avoided. The patients were maintained on mechanical ventilation to keep the EtCO$_2$ at 30–40 mmHg during two-lung ventilation (6–8 ml kg$^{-1}$, 12–14 tpm) and to keep the EtCO$_2$ at 30–50 mmHg

Table 1. Time chart of anesthetic management.

| Premedication | Loading (10 min) | GA Induction | GA Maintenance | GA Recovery |
|---|---|---|---|---|
| Group OFA | Dexmedetomidine (1µg kg$^{-1}$, 4µg/ml) | Dexmedetomidine | Dexmedetomidine * | |
| | | (0.5 µg kg$^{-1}$ h$^{-1}$) | (0.5 µg kg$^{-1}$ h$^{-1}$) | |
| | | Ketorolac | Sevoflurane # | Palonosetron |
| | | (30 mg) | (1%-3%) | 0.25 mg |
| | | Etomidate | Cisatracurium | Neostigmine |
| | | (0.2–0.3 mg kg$^{-1}$) | (2–4 mg per 30 min) | (up to 2 mg) |
| | | Cisatracurium (0.2 mg kg$^{-1}$) | | Atropine (0.2–1 mg) |
| Group OA | | | | |
| | | Sufentanil | Remifentanil # | Namefene [19, 20] |
| | | (0.5 µg kg$^{-1}$) | (200–500 µg h$^{-1}$) | 0.5 mg |
| | | Etomidate | Sevoflurane # | Palonosetron |
| | | (0.2–0.3 mg kg$^{-1}$) | (1%-3%) | (0.25 mg) |
| | | Cisatracurium | Cisatracurium | Neostigmine |
| | | (0.2 mg kg$^{-1}$) | (2–4 mg per 30 min) | (up to 2 mg) Atropine (0.2–1 mg) |

* Infusion was stopped 40 min before end of surgery.

# At end of surgery, inhalation and infusion were terminated.

GA: general anesthesia.

during the period of one-lung ventilation (6 ml kg$^{-1}$, 14–16 tpm) with 100% oxygen inhalation. Intraoperative normothermia was maintained with forced air warming blankets positioned over the exposed parts of the body. Hypotension (MAP < 60 mmHg) was treated with norepinephrine, and bradycardia (HR < 45 bpm) was treated with atropine. Nitroglycerine was administered when MAP > 20% of the preinduction baseline values. The target of PTI was from 40 to 60 and that of WLI ranged from 40 to 60. The tracheal extubation was left to the clinician's discretion when patients achieved a regular spontaneous breathing pattern.

All patients were placed in a lateral position with operation side up after anesthesia induction. After surgical disinfection, in-plane ultrasound-guided (Voluson i, GE USA) single-injecting thoracic paravertebral blockade (TPVB) at T5-6 was performed. Once the image of the disposable injection needle (18G 10 cm, Kangdelai China) was caught in a site between the internal intercostal membrane and the pleura, ropivacaine (0.5%, 15 ml) was administered over 30s. During injection, an anterior displacement of the pleura (downward movement) was observed.

## Outcomes

The primary outcome of this study was intraoperative PTI reading recorded at after entering the operating room (PTI0, baseline value), 10min after infusion of loading dose dexmedetomidine (PTI1), after intubation (PTI2), 30min after one-lung ventilation (PTI3), recovery of double lungs ventilation (PTI4), 5min after extubation in the operation room (PTI5). Secondary outcomes include WLI reading, MAP and HR at same timepoints as PTI, arterial partial pressure of oxygen, blood glucose concentration and lactic acid value from intraoperative blood gas at after entering the operating room (BGA0, baseline value), 1h after surgery begun (BGA1), 2h after surgery begun (BGA2), 5min after extubation in the operation room (BGA3). Total consumption of anesthesia medications, time to passage of flatus (the time from the end of operation to the patient complaining of the first spontaneous breaking wind event), PONV, length of stay, pH and SpO$_2$ were measured and recorded.

In our trial registered with Clinical-Trials.gov, PI (pain index) was declared as primary outcome, and potassium concentration was recorded as secondary outcome. PI is a value indicating the degree of pain in awake subjects and is not suitable for patients under anesthesia. Therefore, our study did not report PI as primary outcome. Since some patients received intravenous potassium supplementation due to BGA0 blood gas detection of hypokalemia before induction of anesthesia, the intraoperative blood potassium values were not recorded.

## Power analysis and sample size calculation

The sample size requirement was based on preliminary data from a previous pilot study of 15 patients undergoing VATS with OA which mean (*s*) PTI was 55.5 (10.4) after open chest. We assumed that OFA could achieve an equal PTI. The significance level is 0.05, the power is 0.9, the true difference is 6 and the standard deviation is 10. We calculated that each group should include 44 patients (PASS 11). Considering that the postoperative follow-up was carried out in the ward before discharge, the dropout rate was determined to be 10%, We recruited 50 patients in each group.

## Statistical analysis

The categorical data were analyzed using the chi-squared test for independence or the Fisher's exact test. The quantitative data were analyzed using the unpaired Student's *t*-test for significance. If the data were not normally distributed as assessed by the Shapiro–Wilk test, the

Wilcoxon rank-sum test was used. Descriptive parameters are represented as means (SD). Highly skewed quantitative data are presented as median with interquartile range (IQR).

Repeated measures analysis of variance and student t-test were used to compare PTI, WLI, MAP, HR at six time points and $PaO_2$, lactic acid levels and blood glucose concentration at four time points within and between groups. Statistical significance was set at $P < 0.05$. All statistical analyses were performed using the SPSS 17.0.

## Results

A total of 222 patients who from November 2018 to March 2019 were assessed for eligibility, finally 100 patients were enrolled in this clinical investigation and were comparable in the two groups (Fig 1). Three patients were excluded due to discontinued intervention: 1 case turned to open surgery, 1 peripheral venous catheter slid out during surgery, 1 patient needed re-exploration for postoperative intra-thoracic bleeding on postoperative day 1. Outcomes of

### CONSORT 2010 Flow Diagram

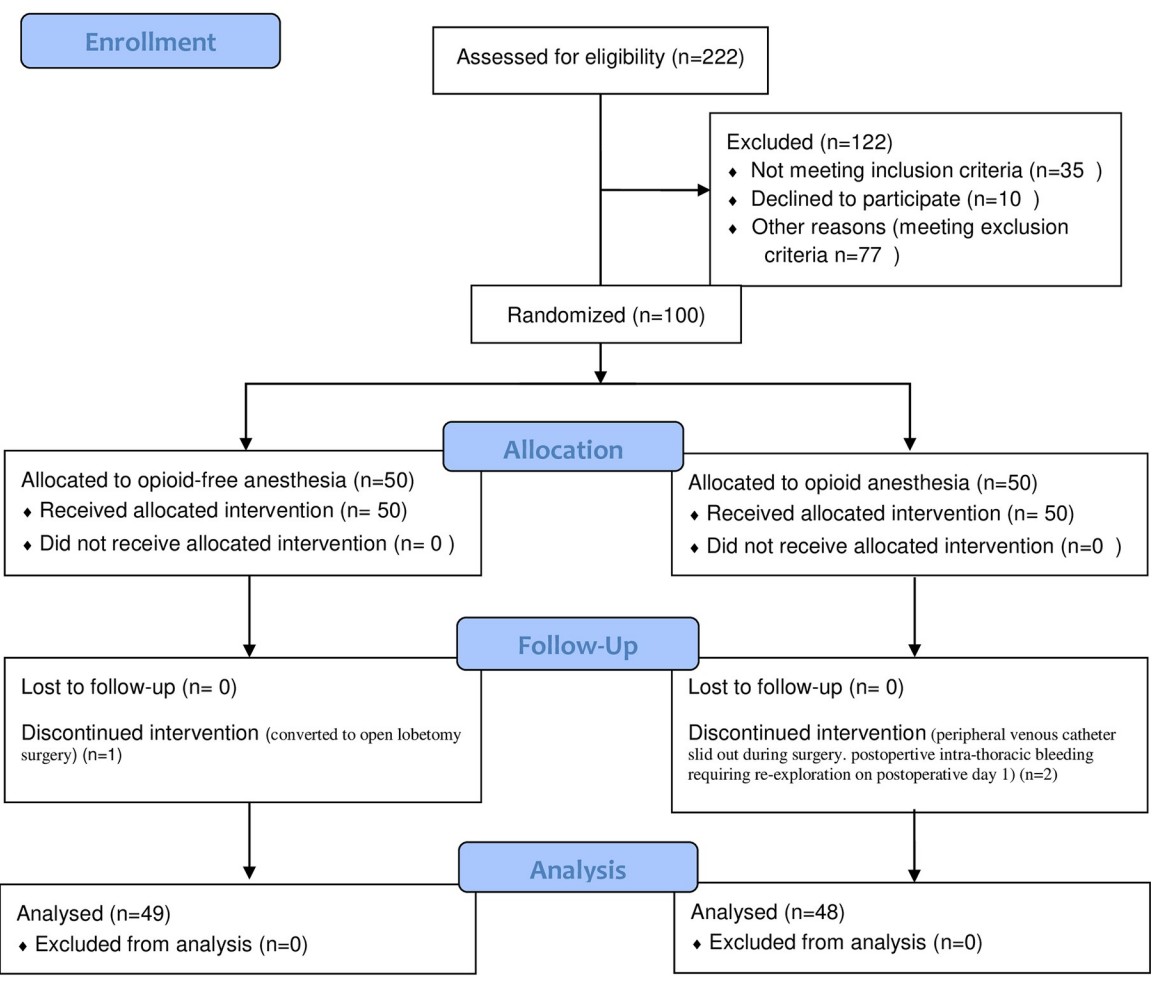

**Fig 1. Study flow diagram.**

**Table 2. Clinical characteristics.**

| | Group OFA | Group OA | P–value |
|---|---|---|---|
| | n = 49 | n = 48 | |
| Gender (Female/Male) | 27/22 | 29/19 | 0.68[#] |
| Age (IQR) | 55 (47.5–60.5) | 55 (50.3–59.3) | 0.93[*] |
| Body Mass Index (BMI, IQR) | 23.4 (21.3–25.5) | 23.4 (22.2–26.4) | 0.27[*] |
| Sites of VATS (Left/Right, n) | 14/35 | 15/33 | 0.83[#] |
| Wedge resection/ Lobectomy/both: n | 7/40/2 | 10/38/0 | 0.28[#] |
| Duration of anesthesia (SD, min) | 151.3 (47.6) | 160.8 (68.6) | 0.44[$] |
| Duration of Surgery (SD, min) | 137.1 (47.2) | 144.6 (68.9) | 0.54[$] |
| Fluid Infusion Volumes (IQR, ml) | 1000 (1000–1250) | 1100 (937.5–1263) | 0.39[*] |
| Urinary volume (IQR, ml) | 350 (250–500) | 300 (225–500) | 0.89[*] |
| Dexmedetomidine consumption (IQR, $\mu$g) | 140 (130–165) | — | |
| Remifentanil consumption (IQR, mg) | — | 1.3 (1.0–1.6) | |
| End-expiratory sevoflurane concentration at end of surgery (IQR, %) | 1.4 (1.2–1.7) | 1.3 (1.2–1.4) | 0.15[*] |
| Ephedrine consumption (IQR, mg) | 0 (0–6.0) | 0 (0–5.3) | 0.47[*] |
| Norepinephrine consumption (IQR, $\mu$g) | 8 (0–46) | 22 (0–85) | 0.11[*] |
| Recovery time (SD, min) | 13.2 (6.1) | 9.5 (4.1) | < 0.001[$] |
| Extubation time (SD, min) | 12.5 (5.7) | 9.7 (4.1) | 0.007[$] |
| Remove drainage tube time (IQR, day) | 4 (3.5–6) | 5 (4–6) | 0.62[*] |
| Dizziness | 2 (4.3%) | 12 (25%) | 0.005[#] |
| Postoperative nausea and vomiting | 2 (4.3%) | 20 (41.7%) | <0.001[#] |
| Time to passage of flatus (IQR, day) | 1 (1–1) | 1 (1–2) | 0.02[*] |
| Hospital Discharge time (95%CI, day) | 7.2(4.2) | 6.6(1.7) | 0.66[*] |

Data are presented as relative number of patients, mean ± standard deviation (SD), median (IQR). Times are shown in minutes.

[*] Wilcoxon rank-sum test

[#] Fisher's exact test

[$] Student's *t*-test.

Recovery time: time from intravenous injection of neostigmine and/or nalmefene to WLI >90; Extubation time: time from the completion of intravenous neostigmine and/or nalmefene to the removal of the tracheal tube; Time to passage of flatus: the time from the end of operation to the first spontaneous breaking wind event. If the duration is less than 24h, it is recorded 1 day; 24h-48h is 2 days, 48h-72h is 3 days, and so on.

these patients were not recorded in the postoperative period. The remaining 97 patients were suitable for analysis (Fig 1). The two groups had similar characteristics at baseline (Table 2).

The results of single factor (time) repeated measurement analysis showed that PTI value, WLI value, PaO$_2$, lactic acid level and blood glucose concentration of the two groups showed significant changes at different time points ($P < 0.001$). At the same time, the linear fit test of the trend of PTI, WLI, PaO$_2$, lactic acid and blood glucose under either opioid-free anesthesia or opioid-based anesthesia were also statistically significant (Tables 3 and 4, Figs 2 and 3, $P <0.001$).

PTI readings between two groups were not significantly different at six time-points during the surgery ($P > 0.05$, Fig 2). The WLI readings at PTI1, PTI2, PTI3 and PTI4 were statistically significant lower in group OFA than those in group OA (such as 46.7 at PTI2 in group OFA vs. 54.5 at PTI2 in group OA; difference, 9.5 [95% CI, 3 to 13]; $P < 0.001$), and there was also the interaction between grouping factors * time factors ($P <0.001$).

The results of single factor (time) repeated measurement analysis showed that (Table 3, Fig 3) the blood glucose levels of patients without opioid anesthesia were significantly increased (20%) at BGA1, BGA2 and BGA3 ($P < 0.01$) which can be considered as blood glucose levels

**Table 3. Summary of PTI and WLI with repeated measurement (mean, 95% CI).**

| | Group | PTI0 | PTI1 | PTI2 | PTI3 | PTI4 | PTI5 | P | | |
|---|---|---|---|---|---|---|---|---|---|---|
| | | | | | | | | Time | Group | Time*Group |
| PTI | OFA | 91.8 (89.5–94.1) | 88.2 (84.7–91.8) | 61.9 (58.6–65.2) # | 53.6 (50.0–57.3) # | 59.7 (55.7–63.8) # | 92.7 (90.8–94.6) | < 0.001* | 0.86 | 0.62 |
| | OA | 91.3 (89.1–93.5) | 88.8 (85.2–92.4) | 63.0 (59.6–66.4) # | 54.2 (50.1–57.8) # | 60.6 (56.6–64.5) # | 88.8 (84.7–92.9) | | | |
| WLI | OFA | 96.3 (95.0–97.5) | 82.7 (77.8–8.5) #^ | 46.9 (42.8–51.1) #^ | 47.8 (44.3–51.3) #^ | 55.2 (51.4–59.0) #^ | 88.4 (86.9–89.9) # | < 0.001* | < 0.001 | < 0.001 |
| | OA | 95.4 (93.9–96.9) | 90.9 (86.6–95.2) | 54.5 (51.0–58.3) # | 53.2 (50.6–55.8) # | 59.5 (56.0–63.0) # | 88.6 (87.2–89.9) | | | |
| MAP | OFA | 106 (102–109) | 107 (101–111) | 115 (110–120) ^ | 88 (84 (92) # | 89 (85–93) # | 98 (94–102) ^ | < 0.001* | 0.89 | <0.001 |
| | OA | 104 (101–108) | 102 (99–106) | 105 (101–110) | 86 (83–89) # | 93 (90–97) # | 109 (105–113) | | | |
| HR | OFA | 73 (69–76) | 71 (68–74) | 78 (74–81) | 73 (70–76) | 70 (66–73) | 74 (71–76) ^ | < 0.001 | 0.004 | 0.001 |
| | OA | 76 (72–80) | 75 (71–78) | 82 (77–86) | 74 (71–77) | 74 (71–76) | 87 (82–92) # | | | |

PTI0: baseline value, PTI1: 10min after infusion of loading does dexmedetomidine, PTI2: after intubation, PTI3: 30min after one-lung ventilation, PTI4: recovery of double lungs ventilation, PTI5: 5 min after extubation.

WLI: wavelet index; PTI: pain threshold index; MAP: mean arterial pressure; HR: heart rate.

*: Linear fitting test (time) $P < 0.05$

#: Compared to PTI0 $P < 0.05$

^: Compared between two groups $P < 0.05$.

increased during the operation. There was no significant difference in blood glucose levels at different timepoints in group OA ($P > 0.05$). The results of multivariate repeated measure analysis of variance showed that the difference in blood glucose levels between the two anesthesia methods was statistically significant ($P < 0.001$).

The recovery time and extubation time of the OFA group were significantly prolonged ($P < 0.05$). There were not significantly difference in $PaO_2$, pH and $SpO_2$ between the two groups during operation. There was no local infections or hematomata at the puncture site 24 h after operation. During the follow-up period, no patient was transferred to the ICU, had respiratory insufficiency, need reintubation, or died. Acute cerebral infarction occurred in 1 case in group OA, and atrial fibrillation occurred in 1 case after operation. No patients in group OFA had hypoxia ($SpO_2 < 90\%$), hypotension or bradycardia requiring treatment.

**Table 4. Summary of PaO₂, lactic and blood glucose with repeated measurement (mean, 95% CI).**

| | Group | BGA0 | BGA1 | BGA2 | BGA3 | P | | |
|---|---|---|---|---|---|---|---|---|
| | | | | | | Time | Group | Time*Group |
| PaO₂ | OFA | 81.7 (78.8–84.5) | 166.9 (137.4–196.3) # | 282.6 (243.2–322.0) #^ | 84.5 (78.3–90.7) | < 0.001* | 0.39 | 0.01 |
| | OA | 81.6 (79.1–84.1) | 190.6 (157.6–223.7) # | 223.9 (188.6–259.2) # | 82.5 (72.2–92.8) | | | |
| Lactic | OFA | 1.2 (1.1–1.3) | 1.2 (1.2–1.3) | 1.2 (1.1–1.3) | 1.4 (1.3–1.5) # | < 0.001* | 0.71 | 0.06 |
| | OA | 1.3 (1.2–1.4) | 1.2 (1.1–1.3) | 1.2 (1.1–1.3) | 1.5 (1.4–1.7) # | | | |
| Blood Glucose | OFA | 5.6 (5.4–5.7) | 6.7 (6.4–7.0) #^ | 6.6 (6.4–6.8) #^ | 6.3 (6.1–6.6) # | < 0.001* | 0.004 | < 0.001 |
| | OA | 5.8 (5.6–5.9) | 5.8 (5.6–6.0) | 5.9 (5.7–6.1) | 6.2 (5.9–6.5) # | | | |

BGA0: baseline value, BGA1: 1h after surgery begun, BGA2: 2h after surgery begun, BGA3: 5 min after extubation.

PaO₂: arterial oxygen partial pressure.

*: Linear fitting test (time) $P < 0.05$

#: Compared to BGA0 $P < 0.05$

^: Compared between two groups $P < 0.05$.

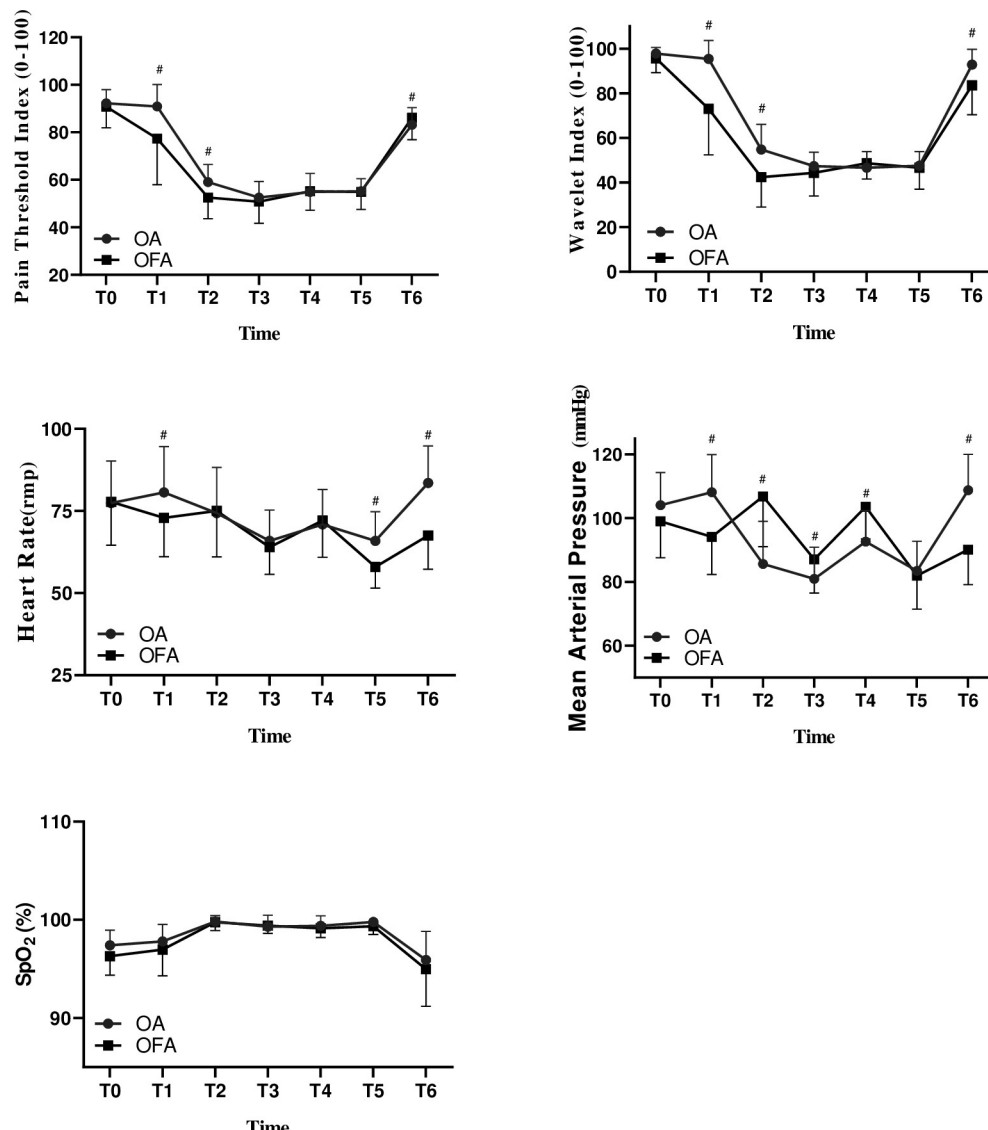

**Fig 2. Changes in wavelet index, pain threshold index, mean arterial pressure and heart rate (mean ±SD) between two groups.** WLI: wavelet index; PTI: pain threshold index; MAP: mean arterial pressure; HR: heart rate. PTI0: baseline value, PTI1: 10min after dexmedetomidine infusion (1μg kg$^{-1}$), PTI2: after intubation, PTI3: 30min after one lung ventilation, PTI4: both lungs ventilation, PTI5: 5 min after extubation. *: $p < 0.05$ between groups.

## Discussion

This prospective randomized study showed that, compared with opioid anesthesia, our OFA program can provide the same sufficient intraoperative analgesia-nociception balance under the guidance of PTI monitoring in patients receiving VATS, for ASA I-II, 18–65 years old, nonobese patients. However, in the OFA group, deeper intraoperative sedation and higher blood glucose were observed.

In the present study, with intraoperative pain management guided by the PTI rather than via hemodynamic indices, the PTI readings in both groups were similar at all the time points and effectively maintained the optimal range from 40 to 60 during surgery, which is thought to predict adequate analgesia to nociceptive stimuli. Compared with OA, there are few reports on

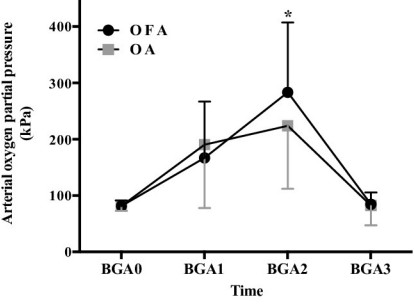 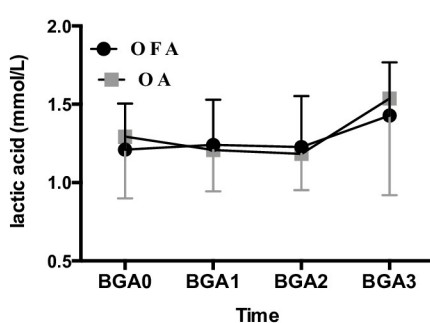

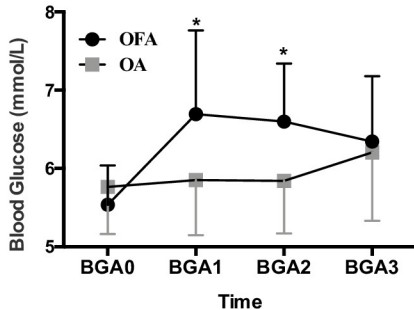

**Fig 3. Changes in arterial oxygen partial pressure, lactic and blood glucose with blood gas (mean±SD) between two groups.** BGA0: baseline value, BGA1: 1h after surgery begun, BGA2: 2h after surgery begun, BGA3: 5 min after extubation. PaO$_2$: arterial oxygen partial pressure. *: $p < 0.05$ between two groups.

the efficacy of intraoperative monitoring of the nociceptive-antinociceptive effects of OFA for patients under general anesthesia. Therefore, the effect of nonopioid anesthesia has been criticized [21]. PTI has become a new parameter for balancing nociception and analgesia during surgery and emphasizes the adequacy of perioperative analgesia in unconscious patients [22–24]. OFA is a multimodal anesthesia associated with hypnotics, N-methyl-D-aspartate antagonists, local anesthetics, anti-inflammatory drugs, and α-2 agonists. Tripathy and colleagues [25] found that patients undergoing magnetic resonance imaging with axillary dissection for breast cancer achieved superior analgesia with opioid-free pectoral nerve block-based anesthesia that involved dexmedetomidine, propofol, and isoflurane. We utilized a specific OFA protocol that included dexmedetomidine, sevoflurane and TPVB to replace opioids [26]. Dexmedetomidine is a highly, potent, selective αlpha 2-adrenergic agonist with intrinsic analgesic properties that can last for at least 12 h after surgery. Due to its analgesic properties, it has been used as an opioid substitute in various surgical interventions [10, 27–29]. A systematic review and meta-analysis from Grape et al. found that intraoperative dexmedetomidine was superior to remifentanil, and its effects on pain can last up to 24 hours after surgery [30]. A loading dose of dexmedetomidine (1 μg kg$^{-1}$) infusion before the induction of anesthesia usually exhibits a central anti-sympathetic effect that may deepen anesthesia and spare the dose of anesthetics, and can provide an optimal balance between adequate postoperative analgesic properties and adverse effects in sensory nerve blocks [31]. Han and colleagues reported that the intraoperative administration of a loading dose of dexmedetomidine (1 μg kg$^{-1}$) combined with sevoflurane (compared to the use of propofol) led to significant stability in blood pressure during the operative period [32]. When compared with systemic agents, regional blockade is most effective for controlling movement-evoked pain [33]. TPVB not only causes dense somatic afferent blockades, but also completely blocks transmission within the

sympathetic chain (due to the anatomy of the paravertebral space). TPVB can produce the same effective analgesia compared to thoracic-epidural-blocks, and can reduce the consumption of opioids without the negative events associated with epidural blocks [34]. In a prospective, randomized, blinded study of patients receiving VATS, the authors observed relief of pain (especially pain during coughing) after TPVB for up to 48 hours. In addition, intravenous infusion of dexmedetomidine significantly increased the duration of intermuscular sulcus brachial plexus block analgesia and reduced cumulative opioid consumption [35, 36].

The WLI is a similar EEG-derived parameter to the BIS and is used to monitor the depth of sedation under GA [22, 37, 38]. Our results showed that with similar PTI readings between the two groups, WLI readings in the OFA group showed a more apparent decrease (10%) than those readings in the OA group. In addition, dexmedetomidine produced a state resembling stage 2 NREM sleep in the EEG with delta and alpha ranges of activity and characteristic arousal sedation [39]. Xu and colleagues [37] reported that, after dexmedetomidine infusion (a loading dose of 0.75 µg kg$^{-1}$ and a continuous infusion of 0.5 µg kg$^{-1}$ h$^{-1}$), the calculated cutoff WLI readings ranged from 55.8–93.7. Yan and Feng reported that injection equivalent analgesia doses of sufentanil (0.2 µg kg$^{-1}$) and remifentanil (2 µg kg$^{-1}$) produced mildly decreased WLI readings from 84.7–98.6 and 91.4–99.0 respectively [40]. Zhang and colleagues reported that, in patients undergoing posterior lumbar interbody fusion surgery with inhalation sevoflurane ranging from 1.5 vol % to 4 vol %, the WLI readings at the incision were 46.7 and 60 at skin closure [41]. Gu and colleagues [39] reported that due to less propofol consumption, as well as different action sites between dexmedetomidine and propofol, the BIS value for loss consciousness was higher in a loading dose of dexmedetomidine (1µg kg$^{-1}$) with propofol than with an infusion propofol alone. In our study, the consumption of sevoflurane during surgery was not recorded; therefore, we were not sure whether the lower WLI in OFA is caused by the effects of sevoflurane that requires further research. Our results showed that deeper sedation in the non-opioid anesthesia regimen with dexmedetomidine and sevoflurane (compared to opioid-based anesthesia with remifentanil and sevoflurane) was statistically significant during the in period of surgery, and the WLI readings in both groups were maintained at normal values ranging from 40–60.

Our results suggested that in patients under nonopioid anesthesia, the intraoperative blood glucose values increased significantly until the conclusion of surgery. The increase in plasma glucose concentration is one of the main features of the metabolic response to surgery and simultaneously demonstrates the effect of different anesthesia techniques on the stress response [42]. Dexmedetomidine attenuates the intraoperative surgical stress response by decreasing nociceptive transmission and by decreasing cortisol levels, thus leading to lower blood glucose levels [43, 44]. In contrast, dexmedetomidine can excite α2A-adrenoreceptors of pancreatic β-cells and inhibit insulin secretion which may cause hypoinsulinaemia and a resultant state of hyperglycemia [45]. Ghimire and colleagues [46] reported that 0.4 µg kg$^{-1}$ dexmedetomidine decreased plasma insulin levels and mildly increased plasma glucose concentrations (4%) in healthy fasting human subjects. Görges and colleagues [47] reported that elevations in blood glucose levels (of +20%) in children were observed 15 minutes after the induction of anesthesia. Additionally, sevoflurane anesthesia has been shown to increase plasma glucose levels by impairing insulin secretion, inducing insulin resistance, and reducing glucose uptake and utilization [48, 49]. The effects of sevoflurane combined with dexmedetomidine on blood glucose levels and insulin sensitivity in general anesthesia have not yet been reported. In our study, the blood glucose value was determined with the use of arterial blood gases to avoid tissue uptake and glucose. Our results showed that the lactic acid levels were not significantly different between the two groups at the tested time points. The reasons for the increase in blood glucose concentrations in nonopioid anesthesia are that insulin secretion is

inhibited by dexmedetomidine and sevoflurane and/or nonopioid anesthesia cannot effectively control the stress response. These possible mechanisms require future studies.

This study had several limitations. First, intraoperative anesthesiologists cannot be blinded to the study interventions. Second, in our study, single-lumen-bronchial-intubation and BB (instead of a dual-lumen endotracheal tube) were used to reduce the stress response to intubation and extubation. Third, it is impossible to accurately calculate the amount of sevoflurane consumed during an operation; therefore, the end-tidal sevoflurane concentrations of the two groups at the end of operation were compared. Considering that sevoflurane may affect the intraoperative sedation index and analgesia index, we began to calculate the consumption of sevoflurane at different times during the operation in follow-up experiments. Fourth, our research is a study a single-institution, with a small sample size, and modifications have been made to some programs. There are no data reports on PI, since PI can only be used in conscious patients and the serum potassium concentration is not recorded, as some patients are supplemented with potassium, leading to selection bias. Fifth, we could not perform an intention-to-treat analysis including the 3 patients excluded after randomization since we did not follow these patients after exclusion. Finally, some protocol changes were performed. The strict inclusion criteria and other study's limitations may limit the external validity of the study. Further studies will be needed to assess the long-term benefits of the avoidance of opioid use in lung cancer patients, as well as to assess the benefits of increasing survival.

## Conclusions

OFA with dexmedetomidine and sevoflurane plus TPVB seems to be feasible for the management of intraoperative nociception in ASA I-II, 18-65-years old, non-obese patients undergoing VATS for lung cancer. However, a deeper sedation level was produced in OFA than in OA for the achievement of an optimal balance of analgesia-nociception. A significant elevation in blood glucose levels in OFA group was also found. Study's limitations and strict inclusion criteria may limit the external validity of the study, suggesting the need of further randomized trials on the topic.

## Supporting information

**S1 Checklist. CONSORT checklist.**
(DOC)

**S1 File. Study protocol-English.**
(DOCX)

**S2 File. Study protocol-original.**
(DOCX)

**S3 File. Protocol change.**
(DOCX)

## Acknowledgments

The authors would like to thank the operating room teams of the Fourth Hospital of Hebei University, including nurses and surgeons, for their help in permitting the experimental protocol to take place.

## Author Contributions

**Conceptualization:** Yiwen Zhang, Nuoya Chen, Xuelian Zhao.

**Data curation:** Nuoya Chen, Bingsha Zhao.

**Formal analysis:** Yiwen Zhang.

**Investigation:** Guangquan An, Nuoya Chen.

**Methodology:** Yiwen Zhang, Nuoya Chen, Jianfeng Fu, Xuelian Zhao.

**Project administration:** Guangquan An.

**Resources:** Guangquan An.

**Software:** Yiwen Zhang, Bingsha Zhao.

**Supervision:** Jianfeng Fu, Xuelian Zhao.

**Validation:** Jianfeng Fu.

**Writing – original draft:** Guangquan An, Yiwen Zhang, Xuelian Zhao.

**Writing – review & editing:** Xuelian Zhao.

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
