## [Decision Letter · Decision Letter 0]

20 Nov 2020

PONE-D-20-27554

Lower sedation index  and raised blood glucose with opioid-free anesthesia compared to opioid anesthesia for lung cancer patients undergoing video-assisted thoracoscopic surgery: a prospective, randomized, controlled study

PLOS ONE

Dear Dr. Zhao,

Thank you for submitting your manuscript to PLOS ONE. After careful consideration, we feel that it has merit but does not fully meet PLOS ONE’s publication criteria as it currently stands. Therefore, we invite you to submit a revised version of the manuscript that addresses the points raised during the review process.

A rebuttal letter that responds to each point raised by the academic editor and reviewer(s). You should upload this letter as a separate file labeled 'Response to Reviewers'. A marked-up copy of your manuscript that highlights changes made to the original version. You should upload this as a separate file labeled 'Revised Manuscript with Track Changes'.An unmarked version of your revised paper without tracked changes. You should upload this as a separate file labeled 'Manuscript'.

We look forward to receiving your revised manuscript.

Kind regards,

Alessandro Putzu, M.D.

Academic Editor

PLOS ONE

Academic Editoror Comment:

A number of issues have been identified in the review process. While we feel that this manuscript shows promise, we also think that a major revision is needed. Before we can make a decision about this manuscript we want to offer you the opportunity to revise and resubmit the manuscript.

2. Thank you for including your ethics statement: "This prospective, randomized, parallel-group and single-center clinical trial was conducted from November 2018 to March 2019, at Fourth Hospital of Hebei Medical University China. The study complied with the Declaration of Helsinki and was approved by the local ethics committee (Fourth Hospital of Hebei Medical University, Hebei, China, #2018049, Chairperson Prof Guiying Wang, on 20 Nov 2018)."   

 a.Please amend your current ethics statement to include the full name of the ethics committee/institutional review board(s) that approved your specific study.

 b.Once you have amended this/these statement(s) in the Methods section of the manuscript, please add the same text to the “Ethics Statement” field of the submission form (via “Edit Submission”).

3. Please provide additional details regarding participant consent. In the ethics statement in the Methods and online submission information, please ensure that you have specified what type you obtained (for instance, written or verbal, and if verbal, how it was documented and witnessed).

4. In your Methods section, please ensure you explain any deviations from your original protocol.

5. We noted several instances of p < 0.000 in your manuscript. To comply with PLOS ONE submission guidelines, please report exact p-values for all values greater than or equal to 0.001. P-values less than 0.001 may be expressed as p < 0.001. For more information on PLOS ONE's expectations for statistical reporting, please see https://journals.plos.org/plosone/s/submission-guidelines.#loc-statistical-reporting.

6. Thank you for stating the following financial disclosure:

"No"

7. Thank you for stating the following in your Competing Interests section: 

"No"

Reviewers' comments:

Reviewer's Responses to Questions

**Comments to the Author**

1. Is the manuscript technically sound, and do the data support the conclusions?

Reviewer #1: Yes

Reviewer #2: Yes

Reviewer #3: Partly

Reviewer #4: Partly

Reviewer #5: Yes

2. Has the statistical analysis been performed appropriately and rigorously? 

Reviewer #1: Yes

Reviewer #2: Yes

Reviewer #3: Yes

Reviewer #4: No

Reviewer #5: Yes

3. Have the authors made all data underlying the findings in their manuscript fully available?

Reviewer #1: Yes

Reviewer #2: No

Reviewer #3: Yes

Reviewer #4: Yes

Reviewer #5: Yes

4. Is the manuscript presented in an intelligible fashion and written in standard English?

Reviewer #1: Yes

Reviewer #2: Yes

Reviewer #3: Yes

Reviewer #4: No

Reviewer #5: Yes

5. Review Comments to the Author

Reviewer #1: I would like to thank the authors for this RCT comparing OFA to OA in lung cancer surgery. The authors show that OFA is associated to non-inferior analgesia, despite a deeper level of sedation and raised blood glucose.

Introduction

Page 3 L 64: please, detail that OFA provides better analgesia in the postoperative period. Inraoperative analgesia is actually the aim of this paper. This will help not making confusion.

See for reference

Analgesic impact of intra-operative opioids vs. opioid-free anaesthesia: a systematic review and meta-analysis.

Frauenknecht J, Kirkham KR, Jacot-Guillarmod A, Albrecht E.Anaesthesia. 2019 May;74(5):651-662. doi: 10.1111/anae.14582. Epub 2019 Feb 25.PMID: 30802933 Free article.

Page 4 line 86: merge this part to the initial part of introduction; this will keep all the details on OFA together. I also think the authors can shorten this part. They do not need to detail on paravertebral block or ketorolac or Dexmethomidine, but just briefly explain that “OFA encompasses different strategies (including dexmethomidine, ketorolac and so on)…and the use of regional anesthesia to act on different components of pain pathways”. They may cite some articles explaining the topic. For reference:

Opioid free anesthesia: evidence for short and long-term outcome.

Bugada D, Lorini LF, Lavand'homme P.Minerva Anestesiol. 2020 Aug 4. doi: 10.23736/S0375-9393.20.14515-2. Online ahead of print.PMID: 32755088

This is not mandatory but will significantly help the reader and make the reading more fluent.

Methods:

line 112 page 5: I guess thy mean “pregnant”

methods are generally well described: inclusion/exclusion, time frame, approval and registration are ok. Sample calculation is ok, as well.

Please, better detail on allocation and blinding. This is important. How do you assign patients to their group and how you keep the investigators unaware of their assignment? How data collectors register data on awareness and nociception during general anesthesia (I guess they are in the OR) without noticing that the patient is infused with one drug or the other?

Page 8 line 162

How do you maintain Sevoflurane ? Basing on Et%? Sevoflurane may interfere with hypnosis..if sevoflurane dose is not standardized, how can we make sure that different levels of hypnosis between OFA and OA patients are given by the treatment rather than by different sevoflurane concentrations? Please, better detail on this point.

Please, better describe in the text the sampling times used in the study: in the figure captions only it is not enough. This point should be clearer from the very beginning to the reader.

Results:

Page 9 line 193: please, remove the timeframe. It is already in the methods.

Page 12 line 247: significant “difference” , I guess.

How do the authors define time for extubation? Any specific parameter?

Discussion:

My main point: this work challenges the idea that OFA does not provide enough analgesia comparing to OA. Part of the criticism on OFA comes from this wrong assumption, that patients will have pain during surgery because they don’t get opioids. This is false, and this work specifically demonstrate it, with an objective measure of analgesia, and as a primary outcome.

So, please, highlight this point. This is good. Let state that this is the first study giving any evidence that intraoperative analgesia is comparable to OA; this will reinforce the concept that OFAis not denying anything to the patients, as already evident for postoperative analgesia. See for reference

Analgesic impact of intra-operative opioids vs. opioid-free anaesthesia: a systematic review and meta-analysis.

Frauenknecht J, Kirkham KR, Jacot-Guillarmod A, Albrecht E.Anaesthesia. 2019 May;74(5):651-662. doi: 10.1111/anae.14582. Epub 2019 Feb 25.PMID: 30802933 Free article.

Page 13 line 278-284: out of context. I would erase this part.

Why do they not use ketamine. Ketamine is a cornerstone of OFA. Is it because it may interfere with sedation measurement? Please, discuss this point.

Dexmedethomidine, as well as ketamine does, may give “false” or “apparent” levels of sedation because of their action of EEG..the authors state it in the discussion, but then they stop, only citing a reference. Please, better detail this point for the reader, because it is extremely interesting in the global evaluation of the study (see conclusions)

Page 15: line 317-320: not clear what do the authors mean

Line 321-323: why is this a limitation?

Conclusion:

I suggest some changes in the message the authors want to give..I suggest not to merely repeat the results

1- OFA provides comparable analgesia (see before)

2- They measure an higher level of sedation but: a) this is a secondary outcome; b) doses of sevoflurane may be adjusted (in future studies) when deeper sedation should not be achieved c) dexmedethomidine, as well as ketamine does, may give “false” or “apparent” levels of sedation because of their action of EEG. Is the deeper level measured really deeper? Further study with sedation as primary endpoint are needed.

3- Glucose is raised but it is probably because of the drugs and not because of surgical stress. So it does not probably mean that OFA does not reduce stress, but only that we need to stay aware and keep higher attention in diabetic patients.

Reviewer #2: I find the original manuscript interesting and it appears to be methodically correct and has not noticed any significant critical issues.

The title is appropriate to the content, informative, concise and clear.

The article deals with a very interesting topic in a comprehensive way and without structural shortcomings.

The data provided is complete and well described and the statistical analysis is absolutely well developed.

The study's objectives and conclusions are clear.

The references are sufficient as a scientific basis for the type of manuscript.

I recommend to the authors a more detailed description of the paravertebral block (which needle was used? Which ultrasound probe? In plane or out of plane technique?)

Reviewer #3: I read with great interest the study entitled “Lower sedation index and raised blood glucose with opioid-free anesthesia compared to opioid anesthesia for lung cancer patients undergoing video-assisted thoracoscopic surgery: a prospective, randomized, controlled study ».

This is an interesting study because it represents a validation of intraoperative analgesia with OFA in comparison to standard opioid anesthesia. To date, we do not have such data.

My comments are as follows:

• The title must be changed because it does not reflect the objective of the present study. According to authors “In this prospective, randomized trial we tested the primary hypothesis that protocol- intraoperative OFA would provide efficacious intraoperative analgesia compared to the opioid-based GA with PTI monitoring”. The results were that you did not demonstrate any difference between OFA and non-OFA groups in term of PTI.

• I would suggest to use non-opioid balanced general anesthesia instead of opioid free anesthesia throughout the manuscript. In the title, you can use OFA but after do not use this term. OFA is a misleading term because people believe that you don’t treat pain.

• In the introduction section, the authors should better focus on cardiac and thoracic surgery. You have published OFA studies on this topic : Guinot PG, Spitz A, Berthoud V, Ellouze O, Missaoui A, Constandache T, Grosjean S, Radhouani M, Anciaux JB, Parthiot JP, Merle JP, Nowobilski N, Nguyen M, Bouhemad B. Effect of opioid-free anaesthesia on post-operative period in cardiac surgery: a retrospective matched case-control study. BMC Anesthesiol. 2019 Jul 31;19(1):136. doi: 10.1186/s12871-019-0802-y. PMID: 31366330; PMCID: PMC6668113. Bello M, Oger S, Bedon-Carte S, Vielstadte C, Leo F, Zaouter C, Ouattara A. Effect of opioid-free anaesthesia on postoperative epidural ropivacaine requirement after thoracic surgery: A retrospective unmatched case-control study. Anaesth Crit Care Pain Med. 2019 Oct;38(5):499-505. doi: 10.1016/j.accpm.2019.01.013. Epub 2019 Feb 5. PMID: 30731138. These studies are interesting because (even they are retrospective ones) they add data to improve the discussion on your finding. They have been performed on thoracic and mini-invasive cardiac surgery that are quite similar to your population. And they use different OFA protocol. You should discuss your protocol in regard of those published.

• How was managed remifentanil dose during the surgery ?

• When did you perform TPVB? At the beginning or at the end of the surgery? Because if you performed the TPVB at the beginning of the surgery, you cannot exclude an effect of TPVB on your results.

• Table 2 : switch PRI with PTI. Please add definitions of abbreviations in the legend for all tables and figures.

• Even the result on BWI is statically significant, I don’t believe that it is clinically relevant. The values of BWI in the two groups were within the fixed ranges. This effect can be explained by the sedation effect of dexmedetomidine. It would be of interest to have values of end tidal sevoflurane during the surgery (T3 to T5). This is a major issue because you demonstrated a difference of sedation level.

• Discussion section: don’t discuss the validity of the PTI. You should better focus on the effectiveness of your OFA protocol regarding analgesia during the surgery. We do not have one type of OFA protocol. Some groups use lidocaine, other mix lidocaine and dexmedetomidine… You should discuss the bias of using TPVB if you perform this block at the beginning of the surgery. You should discuss how you manage sedative drugs during the surgery, thus the low clinical impact of lower sedation level with OFA. You can use recent publications evaluating the impact of a low vs high level of sedation during anesthesia. The results on blood glucose values are interesting because I observe the same finding at bedside in major surgery being performed with OFA. I also note higher lactate values but you did not demonstrate this because you have studied non major thoracic surgery. You should shorten the corresponding paragraph.

• Limitation section: I do not agree with your assessment “Third, it was impossible to calculate the consumption of sevoflurane during the operation, and thus the end-expiratory sevoflurane concentration at the end of surgery in both groups was compared.” It is possible to have value. Do you have a data base? Are you able to get these values?

• Outcomes (Item 6a) Completely defined pre-specified primary outcome measure including how and when it was assessed.

Incompletely. What was the primary objective? PTI or BWI ? Why did you perform sample size calculation only on PTI?

• Sample size (Item 7a)

How sample size was determined

OK

• Sequence generation (Item 8a)

Method used to generate random allocation sequence

Ok

• Allocation concealment (Item 9)

Mechanism used to implement random allocation sequence (such as sequentially numbered containers), describing any steps taken to conceal the sequence until interventions were assigned

Did you stratify randomization? If yes can you develop?

• Blinding (Item 11a)

If done, who was blinded after assignment to interventions (for example, participants, care providers, those assessing outcomes)

OK

• Outcomes and estimation (Item 17a/b)

For the primary outcome, results for each group, and the estimated effect size and its precision (such as 95% confidence intervals)

OK

• Harms (Items 19)

All important harms or unintended effects in each group

Could you provide data on safety ? bradycardia hypotension hypertension tachycardia…..

• Registration (Item 23)

Registration number and name of trial registry

OK

• Protocol (Item 24)

Where trial protocol can be accessed

OK

• Funding (Item 25)

Sources of funding and other support (such as supply of drugs) and role of funders

NO please provide sources of funding

Reviewer #4: Comments on Manuscript # PONE-D-20-27554 from An and colleagues.

The authors propose a randomized study comparing opioid free anesthesia (OFA) using dexmedetomidine infusion with opioid based anesthesia (OA) using fentanyl and remifentanil infusion in patients scheduled for thoracoscopic lung cancer resection. All the patients received paravertebral block. The authors used intraoperative monitors to assess intraoperative nociception and depth of anesthesia (general anesthesia with sevoflurane).

Introduction and abstract:

- the authors should not use “pain intensity or pain relief” in anesthetized patients but should use the term “intraoperative level of nociception or control of intraoperative nociception”.

- Also the abstract “results” mention T1….T2-T5 ? without any explanation in the methods. Please detail.

- The authors seem to rely on the monitoring they use to assess both anesthesia depth and nociception by analysis of EEG. The measures of all the monitors currently available to assess intraoperative nociception are subject to caution. More details about the monitor used here are mandatory – only one single reference (#17) is not enough.

- Introduction should be re-written. The authors have mixed all the potential benefits of OFA, from reduction of postoperative opioid side effects to chronic pain to cancer recurrence. Please focuse on OFA in the context of the present study i.e. control of intraoperative nociception and stress response to surgery.

Methods:

- The authors mention PTI values… please state if these are the values from the Company or the values from the scientific literature. Add more references about the intraoperative use of this monitor.

- Inclusion and exclusion criteria are well reported

- When was the paravertebral block realized: end of anesthesia induction or end of surgery?

- Provide more details about primary and secondary outcomes e.g. blood gas measuring glucose level, lactic acid….

- More the different times of repeated measures to assess both the primary and secondary outcomes should be detailed here in the methods

- Too bad that the authors who state in introduction that OFA may help to control postoperative pain did not assess pain values and analgesics in patients in the recovery room

- Statistical analysis is provided but power calculation is quite unclear because only mean PTI is reported; please describe the confidence interval or standard deviation.

Results:

- Please add in tables 2 and 3 what is statistically significant from what – not clear for the reader or better provide figures (some figures are already displaying the results of the tables)

Discussion: is confusing partly because your primary and secondary outcomes are not clearly describe in introduction and methods. You want to compare the control of intraoperative nociception, depth of anesthesia and stress response between two techniques of anesthesia (OFA versus OA).

Reviewer #5: Study appears well designed.

Limitations should include single institution study too

Tables are appropriate

Paper has grammatical errors and spacing mistakes between sentences which need to be addressed

6. PLOS authors have the option to publish the peer review history of their article (what does this mean?). If published, this will include your full peer review and any attached files.

Reviewer #1: No

Reviewer #2: **Yes: **Domenico Pietro Santonastaso

Reviewer #3: No

Reviewer #4: No

Reviewer #5: No

---

## [Author Response · Author response to Decision Letter 0]

12 Jan 2021

Dear Editor:

On behalf of my co-authors, we thank you very much for giving us an opportunity to revise our manuscript, we appreciate editor and reviewers very much for their positive and constructive.

Comments and suggestions on our manuscript entitled ‘Lower sedation index and raised blood glucose with opioid-free anesthesia compared to opioid anesthesia for lung cancer patients undergoing video-assisted thoracoscopic surgery: a prospective, randomized, controlled study’(PONE-D-20-27554) are all valuable and immensely helpful for revising and improving our paper, as well as the important guiding significance to our research. We have studied reviewer’s comments carefully and have made revision which marked in yellow in the paper. We have tried our best to revise our manuscript according to the comments.

We would like to express our great appreciation to you and reviewers for comments on our paper. We would be glad to respond to any further questions and comments that you may have. Looking forward to hearing from you.

Thank you and best wishes to New Year.

Yours sincerely,

Xuelian Zhao 

31/12/2020

Here is our response to academic editors.

1. Comments. Please ensure that your manuscript meets PLOS ONE's style requirements, including those for file naming. 

 Author response: We are sorry that in the submitted manuscription, the format of the paper does not meet the requirements of PLOS. We have modified it in accordance with the PLOS ONE style templates.

2. Comments: a. Please amend your current ethics statement to include the full name of the ethics committee/institutional review board(s) that approved your specific study.

Author response: We are sorry for our negligence of the full name of the ethics committee. We have added the full name of the ethics committee in the method section and online submission (Lines: 104-105, marked with yellow). 

3. Comments: Please provide additional details regarding participant consent. In the ethics statement in the Methods and online submission information, please ensure that you have specified what type you obtained (for instance, written or verbal, and if verbal, how it was documented and witnessed).

 Author response: Thank you for pointing this out. We have stated in the method section that the patient can only be accepted to the study after signing the informed consent form (Lines: 110, marked with yellow).

4. Comments: In your Methods section, please ensure you explain any deviations from your original protocol.

Author response: Thank you for pointing this out. We have stated in the method section that the investigator should explain to the patient the difference between the experimental method and the original protocol, and then sign the informed consent form after obtaining the patient’s consent before enrolled into the study (Lines: 109-110, marked in yellow).

5. Comments: We noted several instances of p < 0.000 in your manuscript. To comply with PLOS ONE submission guidelines, please report exact p-values for all values greater than or equal to 0.001. P-values less than 0.001 may be expressed as p < 0.001.

Author response: Thank you for pointing this out. We have made corrections in the results section.

6. Comments: Thank you for stating the following financial disclosure:

"No"

Please clarify the sources of funding (financial or material support) for your study. List the grants or organizations that supported your study, including funding received from your institution.

State what role the funders took in the study. If the funders had no role in your study, please state: “The funders had no role in study design, data collection and analysis, decision to publish, or preparation of the manuscript.”

If any authors received a salary from any of your funders, please state which authors and which funders.

If you did not receive any funding for this study, please state: “The authors received no specific funding for this work.”

Author response: Thank you for point this out. All of authors received no specific funding for this work. We have changed the online submission on our behalf.

7. Comments: Thank you for stating the following in your Competing Interests section: 

"No"

Author response: Thank you for point this out. All of authors received no specific funding for this work. We have changed the online submission on our behalf.

8. Comments: Please include captions for your Supporting Information files at the end of your manuscript, and update any in-text citations to match accordingly. Please see our Supporting Information guidelines for more information: http://journals.plos.org/plosone/s/supporting-information.

 Author response: Thank you for point this out. We have corrected.

List of Responses

Reviewer #1: I would like to thank the authors for this RCT comparing OFA to OA in lung cancer surgery. The authors show that OFA is associated to non-inferior analgesia, despite a deeper level of sedation and raised blood glucose.

Dear Reviewer:

Thank you for your letter and for the comments concerning our manuscript entitled “Lower sedation index and raised blood glucose with opioid-free anesthesia compared to opioid anesthesia for lung cancer patients undergoing video-assisted thoracoscopic surgery: a prospective, randomized, controlled study” (PONE-D-20-27554). Those comments are all valuable and very helpful for revising and improving our paper, as well as the important guiding significance to our research. We have studied the comments carefully and have made corrections which we hope meet with approval. Revised portion were marked in yellow in the paper. The main corrections in the paper and the responses to the reviewer’s comments are as flowing:

Responses to the reviewer’s comments:

1. Comments：Introduction

Page 3 L 64: please, detail that OFA provides better analgesia in the postoperative period. Intraoperative analgesia is actually the aim of this paper. This will help not making confusion.

See for reference

Analgesic impact of intra-operative opioids vs. opioid-free anaesthesia: a systematic review and meta-analysis. Frauenknecht J, Kirkham KR, Jacot-Guillarmod A, Albrecht E.Anaesthesia. 2019 May;74(5):651-662. doi: 10.1111/anae.14582. Epub 2019 Feb 25. PMID: 30802933 Free article.

Author response: We do think this is a good suggestion, which will make paper clearer and well reflect the content of the article. We have rewritten the introduction and described that OFA can improve postoperative analgesia in patients undergoing open thoracic surgery, see Lines 53-59 with yellow labels. 

Thank you very much for the literature, see reference 19 (yellow-marked).

2. Comments：Page 4 line 86: merge this part to the initial part of introduction; this will keep all the details on OFA together. I also think the authors can shorten this part. They do not need to detail on paravertebral block or ketorolac or Dexmethomidine, but just briefly explain that “OFA encompasses different strategies (including dexmethomidine, ketorolac and so on)…and the use of regional anesthesia to act on different components of pain pathways”. They may cite some articles explaining the topic. For reference:

Opioid free anesthesia: evidence for short and long-term outcome.

Bugada D, Lorini LF, Lavand'homme P.Minerva Anestesiol. 2020 Aug 4. doi: 10.23736/S0375-9393.20.14515-2. Online ahead of print.PMID: 32755088

This is not mandatory but will significantly help the reader and make the reading more fluent.

Author response: As suggested by the reviewed, we have deleted the description of anesthetics in OFA strategies in the introduction and put this part into the discussion in detail.

Thank you very much for the literature, see reference 24 (yellow-marked).

3. Comments：Methods:

line 112 page 5: I guess thy mean “pregnant”

Author response: We are very sorry for our incorrect writing and have made correction “pregnant”.

4. Comments：

Please, better detail on allocation and blinding. This is important. How do you assign patients to their group and how you keep the investigators unaware of their assignment? How data collectors register data on awareness and nociception during general anesthesia (I guess they are in the OR) without noticing that the patient is infused with one drug or the other?

Author response: We do think these are good questions. We have rewritten the allocation and blinding in detail lines from 128-137 (yellow-marked). Because the management of OFA and OA are different, it is not possible to blind the anesthesiologist and the recorder during the operation.

5. Comments：Page 8 line 162

How do you maintain Sevoflurane ? Basing on Et%? Sevoflurane may interfere with hypnosis if sevoflurane dose is not standardized, how can we make sure that different levels of hypnosis between OFA and OA patients are given by the treatment rather than by different sevoflurane concentrations? Please, better detail on this point.

Author response: We do think these are good questions. We have added an analysis of the effect of sevoflurane on the sedation index in the discussion section (Lines: 355-357).

Sevoflurane affects the depth of sedation and could be monitored by WLI (Jian-Wen Zhang, Zhi-Gan Lv, Ying Kong, Chong-Fang Han, Bao-Guo Wang. Wavelet and pain rating index for inhalation anesthesia: A randomized controlled trial. World J Clin Cases 2020 November 6; 8(21): 5221-5234; Reference 39, yellow-marked). Sevoflurane is an important anesthetic for our study. The experienced anesthesiologists adjust the sevoflurane according to the operation process and various indicators of the patient. Since the consumption of sevoflurane used for anesthesia cannot be accurately calculated, we only record the exhaled concentration of sevoflurane at the end of the operation, and there is no significant difference (P >0.05). Therefore, we cannot determine whether intraoperative consumption of sevoflurane is one of the reasons for the difference in WLI between the two groups. This is also the limitation of this article, which we have already described in the discussion section. 

6. Comments：Please, better describe in the text the sampling times used in the study: in the figure captions only it is not enough. This point should be clearer from the very beginning to the reader.

Author response: We are deeply sorry for the serious omission and have made correction which we hope will meet with approval (Lines195-201, which marked in yellow). 

7. Comments：Results:

7.1. Page 9 line 193: please, remove the timeframe. It is already in the methods.

Author response: We do think this is a good suggestion, and the timeframe has been deleted from Results.

7.2. Page 12 line 247: significant “difference” , I guess.

Author response: We do think this is a good suggestion. It may be that we did not express it clearly. Line 247 means that there is no statistical difference in blood glucose levels of patients under opioid anesthesia at different time points. We have made corrections.

7.3. How do the authors define time for extubation? Any specific parameter?

Author response: We do think these are good questions. In our study, extubation time refers to the time from intravenous bolus of neostigmine to the removal of the tracheal intubation. The indications for tracheal intubation removal are controlled by the anesthesiologist.

8. Comments：Discussion:

My main point: this work challenges the idea that OFA does not provide enough analgesia comparing to OA. Part of the criticism on OFA comes from this wrong assumption, that patients will have pain during surgery because they don’t get opioids. This is false, and this work specifically demonstrate it, with an objective measure of analgesia, and as a primary outcome.

So, please, highlight this point. This is good. Let state that this is the first study giving any evidence that intraoperative analgesia is comparable to OA; this will reinforce the concept that OFA is not denying anything to the patients, as already evident for postoperative analgesia. See for reference (Analgesic impact of intra-operative opioids vs. opioid-free anaesthesia: a systematic review and meta-analysis. Frauenknecht J, Kirkham KR, Jacot-Guillarmod A, Albrecht E. Anaesthesia. 2019 May;74(5):651-662. doi: 10.1111/anae.14582. Epub 2019 Feb 25. PMID: 30802933 Free article).

Author response: We agree with the reviewer’s assessment. Accordingly, we have rewritten discussion and reiterated that the method of observing PTI readings was used for the first time to achieve real-time monitoring of the depth of analgesia during the operation, which confirmed that there was no difference between opioid anesthesia and opioid anesthesia (Lines: 309-311，319-333).

9. Comments：

9.1. Page 13 line 278-284: out of context. I would erase this part.

Author response: We do think these are good suggestion. We have deleted and rewritten this part (Lines:292-335).

9.2. Why do they not use ketamine. Ketamine is a cornerstone of OFA. Is it because it may interfere with sedation measurement? Please, discuss this point.

Author response: We do think these are good questions. Ketamine is a cornerstone of OFA, but ketamine can cause hallucinations and nightmares after surgery, so it is rarely used in ours hospital. Another important reason is that our hospital cannot order ketamine, there is no ketamine for clinical anesthesia.

9.3. Dexmedethomidine, as well as ketamine does, may give “false” or “apparent” levels of sedation because of their action of EEG.. the authors state it in the discussion, but then they stop, only citing a reference. Please, better detail this point for the reader, because it is extremely interesting in the global evaluation of the study 

Author response: Thank you for pointing this out. We have detail in discussion (Lines:348, 356-359, yellow-marked)

10. Comments：

10.1. Page 15: line 317-320: not clear what do the authors mean

Author response: Thank you for pointing this out. We have rewritten this part, see lines 385-389 for details (yellow-marked).

10.2. Line 321-323: why is this a limitation?

Author response: We do think these are good questions. Because single-lung airway management in VATS generally uses dual-lumen endotracheal catheters, the use of single-lumen endotracheal catheters plus bronchial blocker is rare. Moreover, the stress response of double-lumen endotracheal tube intubation is more severe than that of single-lumen tube.

11. Comments：Conclusion:

I suggest some changes in the message the authors want to give. I suggest not to merely repeat the results

1- OFA provides comparable analgesia (see before)

2- They measure an higher level of sedation but: a) this is a secondary outcome; b) doses of sevoflurane may be adjusted (in future studies) when deeper sedation should not be achieved c) dexmedethomidine, as well as ketamine does, may give “false” or “apparent” levels of sedation because of their action of EEG. Is the deeper level measured really deeper? Further study with sedation as primary endpoint are needed.

3- Glucose is raised but it is probably because of the drugs and not because of surgical stress. So it does not probably mean that OFA does not reduce stress, but only that we need to stay aware and keep higher attention in diabetic patients.

Author response: As suggested by the reviewer, we have rewritten the conclusions. In our OFA regimen, sevoflurane plays an important role. If the intraoperative consumption of sevoflurane can be clarified, it can be more clearly explained whether sevoflurane is the cause of lower sedation index in OFA except dexmedetomidine. Therefore, in the new study, we calculate the consumption of sevoflurane in different time intervals and the number of changing concentrations of sevoflurane during the operation.

Once again, Special thanks to you for your good comments and suggestions.

Reviewer #2: I find the original manuscript interesting and it appears to be methodically correct and has not noticed any significant critical issues.

The title is appropriate to the content, informative, concise and clear.

The article deals with a very interesting topic in a comprehensive way and without structural shortcomings.

The data provided is complete and well described and the statistical analysis is absolutely well developed.

The study's objectives and conclusions are clear.

The references are sufficient as a scientific basis for the type of manuscript.

Dear Reviewer:

Thank you for your letter and for the comments concerning our manuscript entitled “Lower sedation index and raised blood glucose with opioid-free anesthesia compared to opioid anesthesia for lung cancer patients undergoing video-assisted thoracoscopic surgery: a prospective, randomized, controlled study” (PONE-D-20-27554). Your comment is valuable and very helpful for revising and improving our paper, as well as the important guiding significance to our research. We have studied the comments carefully and have made corrections which we hope meet with approval. Revised portion were marked in yellow in the paper. 

Responses to the reviewer’s comments:

1. Comments：I recommend to the authors a more detailed description of the paravertebral block (which needle was used? Which ultrasound probe? In plane or out of plane technique?)

Author response: We do think this is a good suggestion. According to your suggestions, we have perfected the operation of thoracic paraspinal block under ultrasound guidance (Lines: 184-188, marked in yellow).

Once again, Special thanks to you for your good comments and suggestions.

Reviewer #3: I read with great interest the study entitled “Lower sedation index and raised blood glucose with opioid-free anesthesia compared to opioid anesthesia for lung cancer patients undergoing video-assisted thoracoscopic surgery: a prospective, randomized, controlled study ».

This is an interesting study because it represents a validation of intraoperative analgesia with OFA in comparison to standard opioid anesthesia. To date, we do not have such data.

Dear Reviewer:

Thank you for your letter and for the comments concerning our manuscript entitled “Lower sedation index and raised blood glucose with opioid-free anesthesia compared to opioid anesthesia for lung cancer patients undergoing video-assisted thoracoscopic surgery: a prospective, randomized, controlled study” (PONE-D-20-27554). Those comments are all valuable and very helpful for revising and improving our paper, as well as the important guiding significance to our research. We have studied the comments carefully and have made corrections which we hope meet with approval. Revised portion were marked in yellow in the paper. The main corrections in the paper and the responses to the reviewer’s comments are as flowing:

Responses to the reviewer’s comments:

1. Comments：• The title must be changed because it does not reflect the objective of the present study. According to authors “In this prospective, randomized trial we tested the primary hypothesis that protocol- intraoperative OFA would provide efficacious intraoperative analgesia compared to the opioid-based GA with PTI monitoring”. The results were that you did not demonstrate any difference between OFA and non-OFA groups in term of PTI.

Author response: We do think this is a good suggestion, which will make the title clearer and well reflect the content of the article. We have modified title to “The equal analgesic index and lower sedation index with opioid-free anesthesia compared to opioid anesthesia for lung cancer patients undergoing video-assisted thoracoscopic surgery: a prospective, randomized, controlled study”.

2. Comments：• I would suggest to use non-opioid balanced general anesthesia instead of opioid free anesthesia throughout the manuscript. In the title, you can use OFA but after do not use this term. OFA is a misleading term because people believe that you don’t treat pain.

Author response: We do think this is a good suggestion, which will help us make the paper more rigorous. 

It can be seen from literatures that there are ‘non-opioid anesthesia’ and ‘opioid-sparing anesthesia’ similarly to ‘opioid-free anesthesia’. Currently what can be accepted so far is ‘opioid-free anesthesia’. Opioid-free anesthesia (OFA) is an emerging technique summarized as “multimodal anesthesia” to supply for OBA. As one drug will not replace opioids, it is the association of drugs and/or techniques that allows a good quality general anesthesia with no need for opioids. A rational strategy implies: 1) combining anti-nociceptive agents to target different circuits involved in nociceptive transmission, 2) monitoring nociception, 3) explicitly using sedative effects of anti-nociceptive agents to reduce the doses of hypnotic agents and 4) providing prolonged pain control.

Dario BUGADA, et al. Opioid free anesthesia: evidence for short and long-term outcome. [J] Minerva Anesthesiology 2020 Aug 04. 

J. Frauenknecht, K. R. Kirkham, A. Jacot-Guillarmod and E. Albrecht. Analgesic impact of intra-operative opioids vs. opioid-free anaesthesia: a systematic review and meta-analysis.[J] Anaesthesia 2019, 74, 651–662 

HeleneBeloeil. Opioid-free anesthesia. [J]. Best Practice & Research Clinical Anaesthesiology. 2019;33(3): 353-360

Therefore, in the revision of this article, we use OFA as a conceptual term, while some content descriptions we use non-free anesthesia.

3. Comments：• In the introduction section, the authors should better focus on cardiac and thoracic surgery. You have published OFA studies on this topic: Guinot PG, Spitz A, Berthoud V, Ellouze O, Missaoui A, Constandache T, Grosjean S, Radhouani M, Anciaux JB, Parthiot JP, Merle JP, Nowobilski N, Nguyen M, Bouhemad B. Effect of opioid-free anaesthesia on post-operative period in cardiac surgery: a retrospective matched case-control study. BMC Anesthesiol. 2019 Jul 31;19(1):136. doi: 10.1186/s12871-019-0802-y. PMID: 31366330; PMCID: PMC6668113. Bello M, Oger S, Bedon-Carte S, Vielstadte C, Leo F, Zaouter C, Ouattara A. Effect of opioid-free anaesthesia on postoperative epidural ropivacaine requirement after thoracic surgery: A retrospective unmatched case-control study. Anaesth Crit Care Pain Med. 2019 Oct;38(5):499-505. doi: 10.1016/j.accpm.2019.01.013. Epub 2019 Feb 5. PMID: 30731138. These studies are interesting because (even they are retrospective ones) they add data to improve the discussion on your finding. They have been performed on thoracic and mini-invasive cardiac surgery that are quite similar to your population. And they use different OFA protocol. You should discuss your protocol in regard of those published.

Author response: We have added the suggested content to the manuscript on introduction (Reference 14 and 15, yellow-marked). We have rewritten the introduction and described in detail that OFA can improve postoperative analgesia in patients undergoing open thoracic surgery, see Lines 53-59 with yellow labels.

4. Comments：• How was managed remifentanil dose during the surgery ?

Author response: We do think this is a good question. 

The anesthesia of all patients in this study was completed by the experienced anesthesiologist. The anesthesiologist adjusts the remifentanil infusion rate during the operation according to the progress of the operation, the patient’s blood pressure and heart rate, and his own experience. We recorded the consumption of remifentanil throughout the operation.

5. Comments：• When did you perform TPVB? At the beginning or at the end of the surgery? Because if you performed the TPVB at the beginning of the surgery, you cannot exclude an effect of TPVB on your results.

Author response: We do think this is a good question. 

In our study, after the anesthesia induction, all patients were placed in a lateral position and received single-injecting TPVB guided by an ultrasound machine at T5-6 with ropivacaine (0.5%, 15 ml). In our department of anesthesia, all patients undergoing thoracoscopic lobectomy received general anesthesia plus thoracic paravertebral block anesthesia (before beginning of surgery with different local anesthetic). Because we cannot rule out the analgesic effect of paravertebral block, all patients in this study received paravertebral block with 0.5%ropivacaine 15 ml.

We have discussed effect of TPVB on our result (Lines334-343).

6. Comments：• Table 2 : switch PRI with PTI. Please add definitions of abbreviations in the legend for all tables and figures.

Author response: We are very sorry for the serious omission and have made correction (marked in yellow) which we hope will meet with approval. 

7. Comments: Even the result on BWI is statically significant, I don’t believe that it is clinically relevant. The values of BWI in the two groups were within the fixed ranges. This effect can be explained by the sedation effect of dexmedetomidine. It would be of interest to have values of end tidal sevoflurane during the surgery (T3 to T5). This is a major issue because you demonstrated a difference of sedation level.

Author response: We agree with the reviewer’s assessment.

Our results showed that the BWI readings in group OFA were significantly lower than those of group OA (55.2: 59.5 at T3), however, the BWI readings were in normal depth of sedation. We also made a statement in the discussion section (Lines: 364-365).

8. Comments：• Discussion section: don’t discuss the validity of the PTI. You should better focus on the effectiveness of your OFA protocol regarding analgesia during the surgery. We do not have one type of OFA protocol. Some groups use lidocaine, other mix lidocaine and dexmedetomidine… You should discuss the bias of using TPVB if you perform this block at the beginning of the surgery. You should discuss how you manage sedative drugs during the surgery, thus the low clinical impact of lower sedation level with OFA. You can use recent publications evaluating the impact of a low vs high level of sedation during anesthesia. The results on blood glucose values are interesting because I observe the same finding at bedside in major surgery being performed with OFA. I also note higher lactate values but you did not demonstrate this because you have studied non major thoracic surgery. You should shorten the corresponding paragraph.

Author response: We do think these are excellent suggestions. We have deleted the validity of the PTI and rewritten this part (Lines: 319-343).

Our study results show that there is no significant difference in lactic acid levels at the time points between the two groups, but the lactic acid values after extubation of the two groups are significantly higher than the preoperative basic values.

9. Comments：• Limitation section: I do not agree with your assessment “Third, it was impossible to calculate the consumption of sevoflurane during the operation, and thus the end-expiratory sevoflurane concentration at the end of surgery in both groups was compared.” It is possible to have value. Do you have a data base? Are you able to get these values?

Author response: We do think these are good questions. 

We observed that during the entire period of anesthesia, the senior anesthesiologists need to frequently adjust the sevoflurane inhalation concentration and oxygen flow according to the progress of the operation and the patient's vital signs. Therefore, the amount of sevoflurane cannot be accurately calculated (Consumption of inhaled anesthetics (ml)/hour = constant λ × volatile tank scale (%) × fresh gas flow (L/min). We only record the exhaled concentration of sevoflurane at the end of the operation, and there is no significant difference (P>0.05). Therefore, we cannot determine whether intraoperative consumption of sevoflurane is one of the reasons for the difference in WLI between the two groups. This is one of the limitations of this article. However, sevoflurane is an important anesthetic for our study. which affects the depth of sedation and could be monitored by WLI (Jian-Wen Zhang, Zhi-Gan Lv, Ying Kong, Chong-Fang Han, Bao-Guo Wang. Wavelet and pain rating index for inhalation anesthesia: A randomized controlled trial. World J Clin Cases 2020 November 6; 8(21): 5221-5234). Therefore, in future experiments, we plan to record the number and time of adjusting sevoflurane inhalation concentration in order to calculate the amount of sevoflurane.

10. Comments：• Outcomes (Item 6a) Completely defined pre-specified primary outcome measure including how and when it was assessed.

Incompletely. What was the primary objective? PTI or BWI ? Why did you perform sample size calculation only on PTI?

Author response: Thank you for pointing this out. PTI was the primary objective of our study. The purpose of this study is to observe whether no opioid anesthesia can achieve the equal analgesic effect of traditional opioid anesthesia. Therefore, we set the analgesia index as the primary observation index and calculate the sample size based on the analgesia index.

11. Comments：• Allocation concealment (Item 9)

Mechanism used to implement random allocation sequence (such as sequentially numbered containers), describing any steps taken to conceal the sequence until interventions were assigned

Did you stratify randomization? If yes can you develop?

Author response: We do think this is a good question. We did not perform stratified randomization during random allocation. But in our new study, we are stratified and randomized whether you have diabetes.

12. Comments：• Harms (Items 19)

All important harms or unintended effects in each group

Could you provide data on safety ? bradycardia hypotension hypertension tachycardia…..

Author response: We do think this is a good question. Yes，we recorded the incidence of complications in this experiment, such as sinus brachytherapy, hypertension, hypotension, etc. There was no statistical difference between the two groups.

Once again, Special thanks to you for your good comments and suggestions.

Reviewer #4: Comments on Manuscript # PONE-D-20-27554 from An and colleagues.

The authors propose a randomized study comparing opioid free anesthesia (OFA) using dexmedetomidine infusion with opioid based anesthesia (OA) using fentanyl and remifentanil infusion in patients scheduled for thoracoscopic lung cancer resection. All the patients received paravertebral block. The authors used intraoperative monitors to assess intraoperative nociception and depth of anesthesia (general anesthesia with sevoflurane).

Dear Reviewer:

Thank you for your letter and for the comments concerning our manuscript entitled “Lower sedation index and raised blood glucose with opioid-free anesthesia compared to opioid anesthesia for lung cancer patients undergoing video-assisted thoracoscopic surgery: a prospective, randomized, controlled study”(PONE-D-20-27554). Those comments are all valuable and very helpful for revising and improving our paper, as well as the important guiding significance to our research. We have studied the comments carefully and have made corrections which we hope meet with approval. Revised portion were marked in yellow in the paper. The main corrections in the paper and the responses to the reviewer’s comments are as flowing:

Responses to the reviewer’s comments:

Introduction and abstract:

1. Comments：- the authors should not use “pain intensity or pain relief” in anesthetized patients but should use the term “intraoperative level of nociception or control of intraoperative nociception”.

Author response: We do think this is a good suggestion, which will make our paper clearer and more well-organized. We have made changes in relevant places in the article.

2. Comments：- Also the abstract “results” mention T1….T2-T5 ? without any explanation in the methods. Please detail.

Author response: We are very sorry for the serious omission and have made correction which we hope will meet with approval (Lines194-201, which marked in yellow). 

3. Comments：- The authors seem to rely on the monitoring they use to assess both anesthesia depth and nociception by analysis of EEG. The measures of all the monitors currently available to assess intraoperative nociception are subject to caution. More details about the monitor used here are mandatory – only one single reference (#17) is not enough.

Author response: We do think this is a good suggestion, which will make our paper more credible. We have added three more literatures. (#18: Hao Kong HZ. Research progress of noxious pricking stimulation and analgesic water level monitoring during operation. Journal of Clinical Anesthesiology. 2020;36(06):612-5; #21: Zhen Su LA, Yang Zhang, Lianhua Chen. The clinical value of analgesia index in evaluating the degree of analgesia during general anesthesia. Chinese Journal of Applied Physiology. 2018;34(5):461-3; #22: Qingqing Zhu, Wei Yue, Jinnan Jiang, Miao Zhang,JianxinＹang. Application of target-controlled infusion of remifentanil guided by pain threshold index in laparoscopic subtotal gastrectomy. J Clinical Anesthesiology, 2020;36(7):634-7).

4. Comments：- Introduction should be re-written. The authors have mixed all the potential benefits of OFA, from reduction of postoperative opioid side effects to chronic pain to cancer recurrence. Please focuse on OFA in the context of the present study i.e. control of intraoperative nociception and stress response to surgery.

Author response: We agree the reviewer’s suggestion which will make our paper clearer and well reflect the content of the article. We have rewritten the introduction and described in detail the effects of OFA on intraoperative surgical stress response and nociception in the discussion section.

Methods:

5. Comments：- The authors mention PTI values… please state if these are the values from the Company or the values from the scientific literature. Add more references about the intraoperative use of this monitor.

Author response: We do think this is a good suggestion, which will make our paper more credible. The values of PTI and values of BWI were from the Company（Yibin Wu. Extraction of objective and quantitative indexs of pain, anxiety, depression and other brain function states from electroencephalogram. China Medical Engineering, 2017;25(4):1-7）.

6. Comments：- When was the paravertebral block realized: end of anesthesia induction or end of surgery?

Author response: We are very sorry for the serious omission. After the anesthesia induction, all patients were placed in a lateral position and received single-injecting TPVB. 

7. Comments：- Provide more details about primary and secondary outcomes e.g. blood gas measuring glucose level, lactic acid….

Author response: As the reviewer’s suggestion, we have added and improved the details about primary and secondary outcomes (Lines 194-201 and yellow-marked).

8. Comments：- More the different times of repeated measures to assess both the primary and secondary outcomes should be detailed here in the methods.

Author response: Thank you for pointing this out, which will make our paper more distinct. We have added and improved the details about primary and secondary outcomes (Lines 194-201 and yellow-marked).

9. Comments：- Too bad that the authors who state in introduction that OFA may help to control postoperative pain did not assess pain values and analgesics in patients in the recovery room

Author response: We do think this is a good suggestion, which will make our paper more distinct. Because the post-anesthesia care unit (PACU) of our hospital was rebuilt in 2018, we did not record the patient's PTI readings and BWI readings in the PACU. According to our original protocol, the patients in group OFA were treated with dexmedetomidine plus ketorolac in PCIA for postoperative analgesia and patients in group OA were received dezocine and ketorolac in PCIA for postoperative analgesia. Our results showed that there was no statistical difference in PTI values and BWI values between the two analgesic regimens. Because the postoperative analgesic of the OA group was different from intraoperative opioids, we did not discuss in this article.

10. Comments：- Statistical analysis is provided but power calculation is quite unclear because only mean PTI is reported; please describe the confidence interval or standard deviation.

Author response: We agree with the reviewer’s suggestion, which will make our paper more distinct. We have added standard deviation of PTI.

11. Comments：- Please add in tables 2 and 3 what is statistically significant from what – not clear for the reader or better provide figures (some figures are already displaying the results of the tables)

Author response: As suggested by the reviewer, we have improved the statistically significant of table 2 and table 3 highlighted with yellow marker.

12. Comments：Discussion: is confusing partly because your primary and secondary outcomes are not clearly describe in introduction and methods. You want to compare the control of intraoperative nociception, depth of anesthesia and stress response between two techniques of anesthesia (OFA versus OA).

Author response: We do think this is an excellent suggestion. We have rewritten introduction and methods, to clear that the primary outcome is PTI, and the secondary outcomes are WLI and blood glucose level and discussion part.

Once again, Special thanks to you for your good comments and suggestions.

Reviewer #5: Study appears well designed.

Dear Reviewer:

Thank you for your letter and for the comments concerning our manuscript entitled “Lower sedation index and raised blood glucose with opioid-free anesthesia compared to opioid anesthesia for lung cancer patients undergoing video-assisted thoracoscopic surgery: a prospective, randomized, controlled study” (PONE-D-20-27554). Those comments are all valuable and very helpful for revising and improving our paper, as well as the important guiding significance to our research. We have studied the comments carefully and have made corrections which we hope meet with approval. Revised portion were marked in yellow in the paper. The main corrections in the paper and the responses to the reviewer’s comments are as flowing:

Responses to the reviewer’s comments:

1. Comments: “Limitations should include single institution study too”.

Author response: It is really true as Reviewer suggested that the single institution study is the limitation of our study was added (Lines: 404-405, yellow-marked).

2. Comments: “Paper has grammatical errors and spacing mistakes between sentences which need to be addressed”.

Author response: We are very sorry for our incorrect writing and have made correction.

Once again, Special thanks to you for your good comments and suggestions.

---

## [Decision Letter · Decision Letter 1]

9 Feb 2021

PONE-D-20-27554R1

Equal analgesic index and lower sedation index with opioid-free anesthesia compared to opioid anesthesia for lung cancer patients undergoing video-assisted thoracoscopic surgery: a prospective, randomized, controlled study

PLOS ONE

Dear Dr. Zhao,

Thank you for submitting your manuscript to PLOS ONE. After careful consideration, we feel that it has merit but does not fully meet PLOS ONE’s publication criteria as it currently stands. Therefore, we invite you to submit a revised version of the manuscript that addresses the points raised during the review process.

A number of issues have been identified in the review process. While we feel that this manuscript shows promise, we also think that a major revision is needed. Before we can make a final decision about this manuscript we want to offer you the opportunity to revise and resubmit the manuscript.

We look forward to receiving your revised manuscript.

Kind regards,

Alessandro Putzu, M.D.

Academic Editor

PLOS ONE

Editor Comments:

Major comments

1- The manuscript should be conformed to all points reported in the CONSORT guidelines.

2- Title. I suggest to use a shorter, clearer and objective title, such as: “Opioid-free anesthesia compared to opioid anesthesia for lung cancer patients undergoing video-assisted thoracoscopic surgery: a randomized controlled study”

3- Abstract. The use of T1-2-3-4-5-6 time points are unclear. Readers should be able to understand the time points when reading the abstract. Please increase clarity. I suggest to avoid the use of these acronyms in the abstract.

4- Abstract. You should report that you included only ASA 1-2, 18-65 years old, non-obese, patients.

5- Methods. You registered the protocol at the end of the study (Date of Last Refreshed on：2020/7/5 22:02:01). This is a major limitation.

6- Registration. The study is registered on ChiCTR, not on clincialtrial.gov.

7- Primary outcomes. In the protocol (ChiCTR ) you reported 3 co-primary endpoints: pain evaluation, MAP, analgesic index, pain index. Please report in Methods why you changed the pre-defined outcome. All these outcome should be reported in the manuscript. This is a limitations to report in the discussion.

8- Secondary outcomes. In the protocol you reported as secondary outcomes: exhaust time, PONV, length of stay, PaO2, length of stay, lactic acid, pH, K, SpO2. Please report why you changed this pre-defined outcomes. Multiple outcomes correction should be used.

9- Outcomes. Time points (T1-6) should be consistent between primary and secondary outcomes. For example, timepoint T6 for primary outcome is equal to T4 for secondary outcome. Unclear, should be fixed.

10- Safety outcomes. Please report some major clinical outcomes. This should include in my opinion: mortality, length of hospital stay, ICU admission, reintubation, and respiratory insufficiency.

11- Outcomes. Please specify when and where you assessed pain after extubation (in the PACU? 15 minutes after extubation?)

12- Sample size. You enrolled 100 patients, not 110 patients. 10 patients refused to participate.

13- Sample size. The statement “We recruited 55 patients per group to minimize the chance of insufficient power, in case the observed variability was higher than expected” is incorrect since you enrolled 100 patients.

14- Analysis. An intention to treat analysis (including n=100) should be also performed for at least major clinical outcomes. You should report why you did not use an ITT analysis.

15- All protocol changes should be reported in the manuscript and in the full form in the supplementary material, with reasons (see CONSORT checklist 3b)

16- Methods. Please report which drugs were used or not used, and if allowed in both groups. For example: ketamine, clonidine, dexmedethomidine, magnesium, dexamethasone or corticoids, local anesthetics surgical infiltration, PONV prophylaxis etc.

17-Figure 1. You screened 110 patients. Is it correct that no one met an exclusion criteria? There are errors?

18- Results. Does transfusion volume indicate blood transfusion or crystalloids? Unclear.

19- Discussion and Conclusions. You should report that you included only ASA 1-2, 18-65 years old, non-obese, patients. This do not allow the generalizability of the results.

20- Study limitations should also include: small sample size, change in primary and secondary outcomes.

21- English should be improved.

Minor comments:

1- Abstract, line 3 and 4. In order to improve clarity, it should read: “We assessed whether opioid-free anesthesia would provide effective analgesia-antinociception monitored by analgesia index in video-assisted thoracoscopic surgery.”

2- Abstract, line 11. It should read: “Secondary outcomes were depth of sedation monitoring by wavelet index and blood glucose concentration.”

3- What does “recovery time” and “extubation time” mean? Please improve clarity.

4- Results. Maybe the smoker status should be reported.

Reviewers' comments:

Reviewer's Responses to Questions

**Comments to the Author**

1. If the authors have adequately addressed your comments raised in a previous round of review and you feel that this manuscript is now acceptable for publication, you may indicate that here to bypass the “Comments to the Author” section, enter your conflict of interest statement in the “Confidential to Editor” section, and submit your "Accept" recommendation.

Reviewer #1: All comments have been addressed

Reviewer #3: All comments have been addressed

Reviewer #6: (No Response)

2. Is the manuscript technically sound, and do the data support the conclusions?

Reviewer #1: Yes

Reviewer #3: Yes

Reviewer #6: Yes

3. Has the statistical analysis been performed appropriately and rigorously? 

Reviewer #1: Yes

Reviewer #3: Yes

Reviewer #6: Yes

4. Have the authors made all data underlying the findings in their manuscript fully available?

Reviewer #1: Yes

Reviewer #3: Yes

Reviewer #6: Yes

5. Is the manuscript presented in an intelligible fashion and written in standard English?

Reviewer #1: Yes

Reviewer #3: Yes

Reviewer #6: Yes

6. Review Comments to the Author

Reviewer #1: Thanks for revising the manuscript and answering ti all the questions and doubts. Congratulations for the nice paper.

Reviewer #3: I read with great interest the study entitled “Lower sedation index and raised blood glucose with opioid-free anesthesia compared to opioid anesthesia for lung cancer patients undergoing video-assisted thoracoscopic surgery: a prospective, randomized, controlled study ».

The authors have improved their manuscript by taking in account all comments.

Congrats

Reviewer #6: In the abstract, and throughout, refrain from using p>0.05 and <0.05 or 0.01. Use precise values to 1 or 2 significant figures.

Abstract: extra space in were.

Line 82: “patients were included after the investigators had explained the deviations from the protocol” surely there were no intended deviations from our original protocol? What is meant by this - the study protocol should have been adhered to?

Line 94: was the study design one of non inferiority? If so, what was the non inferiority margin? It looks like the sample size calculation was actually to detect a difference in groups of 6 units of PtI, but I don’t think that was the intention?

Line 164: dose rather than does

Line 169: began rather than begun.

Stats methods: was consideration given to using a mixed model?

What was the adjustment considered for so many tests done? For multiple testing does an adjustment need to be made?

Line 188: use “.” Before Clinical.

Table 1: wedge resection/lobectomy 7/40 is only 47 for OFA. What about the other 2?

And 2 people missing for sites of VATS?

Line 202 talks about repeated measures analysis. This wasn’t mentioned in the stats method section?

7. PLOS authors have the option to publish the peer review history of their article (what does this mean?). If published, this will include your full peer review and any attached files.

Reviewer #1: No

Reviewer #3: No

Reviewer #6: No

---

## [Author Response · Author response to Decision Letter 1]

22 Mar 2021

Here is our response to academic editors.

1. Comments. The manuscript should be conformed to all points reported in the CONSORT guidelines.

Author response: We are sorry that in the submitted manuscription, the format of the paper does not meet the requirements of the CONSORT guidelines. We have modified it in accordance with the CONSORT guidelines.

2. Comments: Title. I suggest to use a shorter, clearer and objective title, such as: “Opioid-free anesthesia compared to opioid anesthesia for lung cancer patients undergoing video-assisted thoracoscopic surgery: a randomized controlled study”.

Author response: We do think this is a good suggestion, which will make title clearer and well reflect the content of the article. We have rewritten the title with yellow labels.

3. Comments: Abstract. The use of T1-2-3-4-5-6 time points are unclear. Readers should be able to understand the time points when reading the abstract. Please increase clarity. I suggest to avoid the use of these acronyms in the abstract.

Author response: We do think this is a good suggestion, which will make our paper clearer and easier to understand. We have made changes in relevant places in the abstract marked with yellow.

4. Comments: Abstract. You should report that you included only ASA 1-2, 18-65 years old, non-obese, patients.

Author response: We do think this is a good suggestion, which will make our paper clearer and more rigorous. We have made changes in relevant places in the abstract marked with yellow.

5. Comments: Methods. You registered the protocol at the end of the study (Date of Last Refreshed on：2020/7/5 22:02:01). This is a major limitation.

Author response: Thank you for pointing this out. Because the original protocol only said to detect arterial blood gas, we added the protocol on 2020/7/5, specifying that the blood gas detection items are PH, PaO2, lactic acid and blood glucose. 

6. Comments: Registration. The study is registered on ChiCTR, not on clincialtrial. gov.

Author response: Thank you for pointing this out. For the clinical registration of this study, I consulted with the Scientific Research Department of our hospital and was informed that both https://www.clinicaltrials.gov/ in the United States and http://www.chictr.org.cn/index.aspx in China are both first-level registration platforms. And finally, we chose ChiCTR.

7. Comments: Primary outcomes. In the protocol (ChiCTR ) you reported 3 co-primary endpoints: pain evaluation, MAP, analgesic index, pain index. Please report in Methods why you changed the pre-defined outcome. All these outcome should be reported in the manuscript. This is a limitations to report in the discussion.

Author response: Thank you for pointing this out. In the end, we use the PTI as the only primary outcomes of this article, because this study mainly observes whether OFA can provide equivalent analgesia compared with traditional OA. According to the study protocol, we also recorded the intraoperative MAP and HR at different time points. We added MAP and HR data in the method and the results depending on your suggestions (Tab 3, Fig 2, marked with yellow). Pain index (PI range 0–100) is an objective and quantitative EEG pain assessment indicator developed. Its corresponding relationship with the VAS scale is as follows: (1) PI <10, no pain, corresponding to VAS scale 0–2; (2) PI = 10–15, pain threshold range, due to different individual pain thresholds, some people can feel pain, some people feel no pain, corresponding to VAS scale 3–4; (3) PI = 16–30, moderate pain, corresponding to VAS scale 5–7; (4) PI >30, severe pain, corresponding to VAS scale 8–10. PI can be used to objectively monitor the degree of pain in conscious patients after surgery. So, our paper does not have data on the pain index.

8. Comments: Secondary outcomes. In the protocol you reported as secondary outcomes: exhaust time, PONV, length of stay, PaO2, length of stay, lactic acid, pH, K, SpO2. Please report why you changed this pre-defined outcomes. Multiple outcomes correction should be used.

 Author response: Thank you for pointing this out. According to the study protocol, we recorded the remove drainage tube time, dizzy, PONV, exhaust time, hospital discharge time, pH, SpO2 and PaO2. We added these items in the method and the results (Tab 2, Tab 4, Fig 3, yellow marked) depending on your suggestions. We found that some patients had hypokalemia before the operation, and this part of the patients received intravenous potassium chloride infusion, so we did not record intraoperative blood potassium value of blood gas.

9. Comments: Outcomes. Time points (T1-6) should be consistent between primary and secondary outcomes. For example, timepoint T6 for primary outcome is equal to T4 for secondary outcome. Unclear, should be fixed.

Author response: Thank you for your suggestion. We have redefined the collection time point of the primary outcomes and the secondary outcomes. The time points of PTI, BWI, MAP and HR were T1-6 (T1: baseline values, T2: 10min after dexmedetomidine infusion (1μg kg-1), T3: after intubation, T4: 30min after one lung ventilation, T5: both lungs ventilation, T6: 5 min after extubation). The time points of blood gas were T1-4 (T1: baseline values, T2: 1h after surgery begun, T3: 2h after surgery begun, T4: 5 min after extubation).

10.Comments：Safety outcomes. Please report some major clinical outcomes. This should include in my opinion: mortality, length of hospital stay, ICU admission, reintubation, and respiratory insufficiency.

Author response: We do think this is a good suggestion, we have added safety results in the results (see Tab 2 and Line272-274, marked with yellow).

11.Comments: Outcomes. Please specify when and where you assessed pain after extubation (in the PACU? 15 minutes after extubation?)

Author response: Sorry for not specifying the time point of T6 and T6. The time point of T6 and T6 are 5 minutes after extubation in the operation room (Lines: 182, 186; yellow marked).

12.Comments: Sample size. You enrolled 100 patients, not 110 patients. 10 patients refused to participate.

Author response: We are deeply sorry for the error and have made correction marked with yellow (Lines: 204-205).

13. Sample size. The statement “We recruited 55 patients per group to minimize the chance of insufficient power, in case the observed variability was higher than expected” is incorrect since you enrolled 100 patients.

Author response: Thank you for pointing this out. This description is indeed inappropriate, and we have deleted this sentence.

14.Comments: Analysis. An intention to treat analysis (including n=100) should be also performed for at least major clinical outcomes. You should report why you did not use an ITT analysis.

Author response: We do think this is good question. The purpose of this study is to test whether the depth of analgesia without opioid anesthesia is equivalent to that of opioid anesthesia. ITT analysis is more suitable for superiority experiments. Therefore, we excluded three patients in accordance with the experimental protocol, and collected data from 97 patients.

15.Comments: All protocol changes should be reported in the manuscript and in the full form in the supplementary material, with reasons (see CONSORT checklist 3b)

Author response: We do think this is good suggestion. Since midazolam may affect the cognitive function of the elderly after surgery, we changed the experimental plan that all patients did not use midazolam before the induction of anesthesia. Before the patient signs the informed consent form, inform the patient of this change, and sign the informed consent form after the patient agrees (Lines: 97-99, marked with yellow). We have added those in the manuscript and in the protocol.

16.Comments: Methods. Please report which drugs were used or not used, and if allowed in both groups. For example: ketamine, clonidine, dexmedethomidine, magnesium, dexamethasone or corticoids, local anesthetics surgical infiltration, PONV prophylaxis etc.

Author response: We do think this is good suggestion. We add Table 1 according to the anesthesia plan and list the anesthetics involved in this study according to the anesthesia steps.

17.Comments: Figure 1. You screened 110 patients. Is it correct that no one met an exclusion criteria? There are errors?

Author response: We do think this is good question. 110 patients met the inclusion criteria and we have made it clear in the article marked with yellow.

18.Comments: Results. Does transfusion volume indicate blood transfusion or crystalloids? Unclear.

Author response: We do think this is good question. No blood transfusion in all patients during the operation. We have modified transfusion volumes in Table 1 to the fluid infusion volumes marked up yellow.

19.Comments: Discussion and Conclusions. You should report that you included only ASA 1-2, 18-65 years old, non-obese, patients. This do not allow the generalizability of the results.

Author response: We do think this is good suggestion, which will make our paper more rigorous. We have added only ASA 1-2, 18-65 years old, non-obese, patients in the discussion and conclusion section (Lines: 281, 396; marked with yellow).

20.Comments: Study limitations should also include: small sample size, change in primary and secondary outcomes.

Author response: We do think this is good suggestion, which will make our paper more rigorous. We have added small sample size, change in primary and secondary outcomes in the limitations (marked up yellow, Lines: 387-389) 

Minor comments:

1.Comments: Abstract, line 3 and 4. In order to improve clarity, it should read: “We assessed whether opioid-free anesthesia would provide effective analgesia-antinociception monitored by analgesia index in video-assisted thoracoscopic surgery.”

Author response: We do think this is good suggestion, which will make our paper more clarity. We have modified it.

2.Comments: Abstract, line 11. It should read: “Secondary outcomes were depth of sedation monitoring by wavelet index and blood glucose concentration.”

Author response: We do think this is good suggestion, we have corrected it.

3.Comments: What does “recovery time” and “extubation time” mean? Please improve clarity.

Author response: We do think this is good question. We have explained the recovery time and extubation time in Table 1 in detail.

4.Comments: Results. Maybe the smoker status should be reported.

Author response: We do think this is good suggestion. Unfortunately, we did not record whether the patient had a history of smoking when collecting the data. When the 5-year follow-up of the patient’s survival, we will include whether the patient has a history of smoking. We appreciate your suggestion.

We should like to express our appreciation to you for suggesting how improve our paper.

List of Responses

Reviewer #6: 

Dear Reviewer:

Thank you for your letter and for the comments concerning our manuscript entitled “Lower sedation index and raised blood glucose with opioid-free anesthesia compared to opioid anesthesia for lung cancer patients undergoing video-assisted thoracoscopic surgery: a prospective, randomized, controlled study” (PONE-D-20-27554). Those comments are all valuable and very helpful for revising and improving our paper, as well as the important guiding significance to our research. We have studied the comments carefully and have made corrections which we hope meet with approval. Revised portion were marked in yellow in the paper. The main corrections in the paper and the responses to the reviewer’s comments are as flowing:

Responses to the reviewer’s comments:

1. Comments：In the abstract, and throughout, refrain from using p>0.05 and <0.05 or 0.01. Use precise values to 1 or 2 significant figures.

Author response: We do think this is a good suggestion and have changed to a more precise p-value (yellow-marked; Lines: 15, 17, 20, 21).

2. Comments：Abstract: extra space in were.

Author response: We are deeply sorry for our incorrect writing and have made correction.

3. Comments：Line 82: “patients were included after the investigators had explained the deviations from the protocol” surely there were no intended deviations from our original protocol? What is meant by this - the study protocol should have been adhered to?

Author response: We do think this is a good question. To avoid the influence of midazolam on cognitive function of the elderly after surgery, we did not use midazolam before general anesthesia induction. Before the patient signs the informed consent form, inform the patient of this change, and sign the informed consent form after the patient agrees. Sorry for the misspelling and we have made correction “pregnant”.

4. Comments：Line 94: was the study design one of non inferiority? If so, what was the non inferiority margin? It looks like the sample size calculation was actually to detect a difference in groups of 6 units of PtI, but I don’t think that was the intention?

Author response: We do think these are good questions. The statistical consultant determined the non-inferiority margin of this study based on the mean and standard deviation of PTI in the pre-experimental, and the intraoperative range of PTI，so the non-inferiority margin for this study was determined to be 6.

5. Comments：Line 164: dose rather than does

Author response: We are sorry for our incorrect writing and have made correction “dose”.

6. Comments：Line 169: began rather than begun.

Author response: We are very sorry for our incorrect writing and have made correction “began.”

7. Comments：Stats methods: was consideration given to using a mixed model?

Author response: We do think this is a good question, we did not consider applying a mixed effects model. The statistical methods used in this study are based on relevant literature and consulting statistics professors.

1. P. Ziemann-Gimmel, A. A. Goldfarb, J. Koppman and R. T. Marema. Opioid-free total intravenous anaesthesia reduces postoperative nausea and vomiting in bariatric surgery beyond triple prophylaxis. British Journal of Anaesthesia, 2014;112: 906–11

2. Uddalak Chattopadhyay, Suchismita Mallik, Sarmila Ghosh, Susmita Bhattacharya, Subrata Bisai, Hirak Biswas. Comparison between propofol and dexmedetomidine on depth of anesthesia: A prospective randomized trial. Journal of Anaesthesiology Clinical Pharmacology; 2014; 30:550-554

3. Kaushic A. THEERTH , Kamath SRIGANESH, K. Madhusudan REDDY, Dhritiman CHAKRABARTI, Ganne S. UMAMAHESWARARAO. Analgesia Nociception Index-guided intraoperative fentanyl consumption and postoperative analgesia in patients receiving scalp block versus incision-site infiltration for craniotomy. Minerva Anestesiologica 2018; 84:1361-1368

8. Comments：What was the adjustment considered for so many tests done? For multiple testing does an adjustment need to be made?

Author response: We do think these are good questions. The PTI, WLI, MAP and HR measurement time points in this study are related to the surgical procedure, and the blood gas collection time point is related to the operation time. Intraoperative adjustments are all adjusted by the anesthesiologist's judgment.

9. Comments：Line 188: use “.” Before Clinical.

Author response: Thank you for pointing these out, we have corrected it.

10. Comments：Table 1: wedge resection/lobectomy 7/40 is only 47 for OFA. What about the other 2? And 2 people missing for sites of VATS?

Author response: Thank you for pointing these out, it is ours mistake. In the OFA group, one patient underwent left lower lobe resection and upper lobe wedge resection, and one patient underwent right upper lobe resection and lower lobe wedge resection, so they are not counted. We have made revisions and re-statistical analysis (see Tab 2, marked-yellow).

11. Comments： Line 202 talks about repeated measures analysis. This wasn’t mentioned in the stats method section?

Author response: We do think these are good questions. We have marked in yellow in the method section “Within and between-group comparisons of either PTI, WLI, MAP and HR at six time points and PaO2, lactic acid levels and blood glucose concentrations at four time points were performed using repeated measurement analysis of variance.”

Once again, Special thanks to you for your good comments and suggestions.

---

## [Decision Letter · Decision Letter 2]

11 Apr 2021

PONE-D-20-27554R2

Opioid-free anesthesia compared to opioid anesthesia for lung cancer patients undergoing video-assisted thoracoscopic surgery: a randomized controlled study

PLOS ONE

Dear Dr. Zhao,

Thank you for submitting your manuscript to PLOS ONE. After careful consideration, we feel that it has merit but does not fully meet PLOS ONE’s publication criteria as it currently stands. Therefore, we invite you to submit a revised version of the manuscript that addresses the points raised during the review process.

We look forward to receiving your revised manuscript.

Kind regards,

Alessandro Putzu, M.D.

Academic Editor

PLOS ONE

Editor Comments:

Thank you for your further work on this manuscript. A number of issues have been identified in the review process. While we feel that this manuscript shows promise, we also think that major revisions are needed. Before we can make a final decision about this manuscript we want to offer you the opportunity to revise and resubmit the manuscript.

Major comments:

1. Methods, page 5, “Eligible patients were included after investigator had explained the deviations from our original protocol and obtained informed consent form signed by the patient.”. What does this sentence mean? Did you explain which kind of deviations to whom? Please improve clarity.

2. Methods, page 6, “Before the patient signs the informed consent form, inform the patient of this change, and sign the informed consent form after the patient agrees.”. What does this sentence mean? What is the change you are talking about? Please improve clarity.

3. Page 6, “Since midazolam may affect the cognitive function of the elderly after surgery, we changed the experimental plan that all patients did not use midazolam before the induction of anesthesia”. This statement is of limited significance and importance since you included patients 18-65 years old

4. Page 6, “To facilitate the emergency treatment of patients during the operation, the anesthesiologists who did not participate in the assessment of the patients at any time were aware of the protocol”. This sentence is unclear and should be improved.

5. Page 6, line 105-106. Details on number of enrolled patients should be reported only in the Results. Please remove this detail from the methods.

6. Table 1. What Namefen is? Was it used only in OA group? Why? Please improve clarity and insert a reference on this drug.

7. Primary outcomes. The protocol reported different primary and secondary outcomes. Please report in methods why you changed the pre-defined outcomes. These are limitations to report in the discussion.

8. Major protocol changes should be reported in the manuscript and in the full form in the supplementary material, with reasons (see CONSORT checklist 3b).

9. Outcomes. Time points (T1-6) are unclear, since are similar for DPI and blood gas analyses outcomes. I suggest to use: PTI0 (=baseline), PTI1…PTI5.For blood gas analysis (BGA) I suggest: BGA0 (=baseline), BGA1…BGA4.

10. You registered the protocol at the end of the study (Date of Last Refreshed on:2020/7/5 22:02:01; date of first version: 2020/7/1 0:47:36). This is a major limitation and must be reported in the Abstract, Methods and Discussion. In the discussion a paragraph should be dedicated to discuss the limitations on protocol registration and change in primary outcomes. The certainty of evidence should be downgraded due to this major limitations. Results should be discussed as exploratory. Conclusions should be tempered.

11. Authors decided not to employ a ITT analysis. I suggest to insert ITT data on safety (mortality, postoperative complications). The lack of an ITT analysis is a major limitation that should be reported in the discussion.

12. Statistical analysis. What was the adjustment considered for so many tests done? For multiple testing does an adjustment need to be made?

13. Results and CONSORT flow diagram. You said that 110 patients met inclusion criteria, but you assessed for eligibility much more than 110 patients. How many patients met exclusion criteria? This data should be reported (Not meeting inclusion criteria n=X, Declined to participate n=X, Other reasons n=X ).

14. What is exhaust time? Please improve clarity.

15. The English must be improved. I suggest to ask for professional help.

Minor comments:

1. Page 5, “The study was registered with ClinicalTrials.gov (ChiCTR 1800019479).”. It should read: “The study was registered on Chinese Clinical Trial Registry (ChiCTR 1800019479).”.

2. Please expand TPVB the first time you used it.

3. Methods. The “Power analysis and sample size calculation” paragraph should be moved below, before statistical analysis paragraph.

4. “dizzy” should probably reads “dizziness”

5. CONSORT diagram. You should report reason of discontinuation of intervention.

Reviewers' comments:

Reviewer's Responses to Questions

**Comments to the Author**

1. If the authors have adequately addressed your comments raised in a previous round of review and you feel that this manuscript is now acceptable for publication, you may indicate that here to bypass the “Comments to the Author” section, enter your conflict of interest statement in the “Confidential to Editor” section, and submit your "Accept" recommendation.

Reviewer #1: All comments have been addressed

Reviewer #3: All comments have been addressed

Reviewer #6: (No Response)

2. Is the manuscript technically sound, and do the data support the conclusions?

Reviewer #1: Yes

Reviewer #3: Yes

Reviewer #6: Yes

3. Has the statistical analysis been performed appropriately and rigorously? 

Reviewer #1: I Don't Know

Reviewer #3: Yes

Reviewer #6: Yes

4. Have the authors made all data underlying the findings in their manuscript fully available?

Reviewer #1: Yes

Reviewer #3: Yes

Reviewer #6: Yes

5. Is the manuscript presented in an intelligible fashion and written in standard English?

Reviewer #1: Yes

Reviewer #3: Yes

Reviewer #6: Yes

6. Review Comments to the Author

Reviewer #1: Thanks to the authors for their replies to all the comments. They have addressed all the issues and the manuscript in now much improved.

Reviewer #3: The authors have addressed all comments of the reviewers 5 and 6. I do not have supplementary comments.

Reviewer #6: Thanks for the revisions to the paper. Many improvements were made, but there are still some points to clarify.

1. Pg 6. Line 100 (track changed version) I feel the statistical sample size calculation text in the manuscript does not adequately match with the words in the response to reviewers.

2. There is now more of a mismatch between 88 pts and 100 pts. Please explain.

3. The 10 patients who refused to participate - did they do so before or after randomisation? If so, say so, and specify the usual time frame between registration/ agreeing to participate and randomisation. This process needs further clarity.

7. PLOS authors have the option to publish the peer review history of their article (what does this mean?). If published, this will include your full peer review and any attached files.

Reviewer #1: No

Reviewer #3: No

Reviewer #6: No

---

## [Author Response · Author response to Decision Letter 2]

11 May 2021

Here is our response to academic editors.

Major comments:

1. Comments. Methods, page 5, “Eligible patients were included after investigator had explained the deviations from our original protocol and obtained informed consent form signed by the patient.”. What does this sentence mean? Did you explain which kind of deviations to whom? Please improve clarity.

Author response: Thank you for point this out which will make paper clearer. We have rewritten the participant eligibility and consent of method, see Lines 86-89 with yellow labels.

2. Comments. Methods, page 6, “Before the patient signs the informed consent form, inform the patient of this change, and sign the informed consent form after the patient agrees.”. What does this sentence mean? What is the change you are talking about? Please improve clarity.

Author response: Thank you for point this out which will make paper clearer. We have rewritten the participant eligibility and consent of method, see Lines 86-89 with yellow labels.

3. Comments. Page 6, “Since midazolam may affect the cognitive function of the elderly after surgery, we changed the experimental plan that all patients did not use midazolam before the induction of anesthesia”. This statement is of limited significance and importance since you included patients 18-65 years old

Author response: Thank you for point this out. It has been reported that midazolam is related to postoperative delirium. Our clinical observations showed that even if it is not for elderly patients with delirium after thoracic surgery, the effect of midazolam cannot be ruled out. Therefore, midazolam is no longer used before anesthesia in patients undergoing thoracic surgery in our hospital. We have rewritten the participant eligibility and consent of method, see Lines 86-89 with yellow labels.

4. Comments. Page 6, “To facilitate the emergency treatment of patients during the operation, the anesthesiologists who did not participate in the assessment of the patients at any time were aware of the protocol”. This sentence is unclear and should be improved.

Author response: Thank you for pointing this out. We have rewritten the sentence, see Lines 114 with yellow labels.

5. Comments. Page 6, line 105-106. Details on number of enrolled patients should be reported only in the Results. Please remove this detail from the methods.

Author response: We do think these are excellent suggestions. We have deleted the details on number of enrolled patients (Lines: 109-110).

6. Comments. Table 1. What Namefen is? Was it used only in OA group? Why? Please improve clarity and insert a reference on this drug.

Author response: We do think this is a good question.

Nalmefene is a mu, kappa, alpha opioid receptor blocker, especially has a strong affinity for mu receptors. Nalmefene has a long action time, high bioavailability and few side effects. Clinically, nalmefene can inhibit or reverse the respiratory depression, sedation and hypotension effects of opioids (Chen Yingqi, Yue Yun, Qing Enming, et al. Nalmefene antagonizes the effect of opioid postoperative respiratory depression-a multicenter, randomized, double-blind, positive drug control study. Chinese Journal of Anesthesiology, 2011,31(3):307-309)

Jia reported that low-dose nalmefene prevents remifentanil-induced postoperative hyperalgesia in patients undergoing gynecological laparoscopic surgery under general anesthesia (Jia Zhen, Chen Yi, Zhang Linlin, et al. The effect of different low-dose nalmefene in preventing postoperative hyperalgesia induced by remifentanil. Chinese Journal of Anesthesiology, 2017, 37(10): 1159-1162.)

Yes, only patients in group OA received nalmefene to reverse the respiratory depression of opioids. We have insert reference on nalmefene in table 1.

7. Comments. Primary outcomes. The protocol reported different primary and secondary outcomes. Please report in methods why you changed the pre-defined outcomes. These are limitations to report in the discussion.

Author response: Thank you for pointing this out. We added these items in the method and the discussion (Lines:190-195, 398-401, yellow marked) depending on your suggestions.

8. Comments. Major protocol changes should be reported in the manuscript and in the full form in the supplementary material, with reasons (see CONSORT checklist 3b).

Author response: We do think this is good suggestion. Since midazolam midazolam is related to the occurrence of delirium in patients after surgery, the administration of midazolam before anesthesia was cancelled. The eligible patient was informed that they aren’t received midazolam and signed a written informed consent form after obtaining the consent of the participant (Lines: 86-89, marked with yellow). We have added those in the method.

9. Comments. Outcomes. Time points (T1-6) are unclear, since are similar for DPI and blood gas analyses outcomes. I suggest to use: PTI0 (=baseline), PTI1…PTI5.For blood gas analysis (BGA) I suggest: BGA0 (=baseline), BGA1…BGA4.

Author response: We do think this is good suggestion, which will make our paper clearer and easier to understand. We have made changes in relevant places in the manuscript marked with yellow.

10. Comments. You registered the protocol at the end of the study (Date of Last Refreshed on:2020/7/5 22:02:01; date of first version: 2020/7/1 0:47:36). This is a major limitation and must be reported in the Abstract, Methods and Discussion. In the discussion a paragraph should be dedicated to discuss the limitations on protocol registration and change in primary outcomes. The certainty of evidence should be downgraded due to this major limitations. Results should be discussed as exploratory. Conclusions should be tempered.

Author response: Thank you for pointing this out. Because the original protocol only said to detect arterial blood gas, we added the protocol on 2020/7/5, specifying that the blood gas detection items are PH, PaO2, lactic acid and blood glucose.

We only applied to modify the protocol once, that is, we submitted an application to modify the protocol at 0:7 on 1/7/2020, and the webmaster replied to agree to the modification at 0:47 on 1/7/2020. After seeing your question, I checked the record, and indeed there were two revision records on 1/7/2020 and 5/7/2020, which made us very confused. After we contacted the webmaster, he replied that it might be a mistake in the website record, because the contents of the two records are exactly the same and the reason maybe the website passed our revision on 5/7/2020.

We have added change in primary and secondary outcomes in the limitations (marked up yellow, Lines: 398-403) 

11. Comments. Authors decided not to employ a ITT analysis. I suggest to insert ITT data on safety (mortality, postoperative complications). The lack of an ITT analysis is a major limitation that should be reported in the discussion.

Author response: We do think this is good suggestion. The ITT analysis included all randomized patients to fully observe the effect of the intervention. Unfortunately, we did not follow up the excluded patients. We have added the limitation of lack of the ITT analysis in the discussion (Lines: 401-403, marked with yellow).

12. Comments. Statistical analysis. What was the adjustment considered for so many tests done? For multiple testing does an adjustment need to be made?

Author response: We do think this is good question. In our study, we have just compared the PTI value at different time points (PTI1-PTI5) to the PTI basic value at PTI0 with ANOVA, and the PTI values in OFA group to those in OA group value at the same time point with t-test, we did not make adjustment. As multiple testing, we only focus on the difference between the two groups, so we did not perform adjustment such as Bonferroni correction or FDR correction.

13. Comments. Results and CONSORT flow diagram. You said that 110 patients met inclusion criteria, but you assessed for eligibility much more than 110 patients. How many patients met exclusion criteria? This data should be reported (Not meeting inclusion criteria n=X, Declined to participate n=X, Other reasons n=X ).

Author response: We do think this is good suggestion. From November 2018 to March 2019, we included 222 patients who underwent thoracoscopic lobectomy. Among them, 35 were due to age > 65 years, 10 were due to BMI > 30%, and 5 had history of alcohol, 62 people had cardiopulmonary insufficiency, and 10 people refused to participate in the experiment. Eventually 100 participants entered the experiment. We have modified the relevant content, see Fig 1 and Lines 216.

14. Comments. What is exhaust time? Please improve clarity.

Author response: We do think this is good question. In the ward, the patient was asked the time from returning to the ward to the first fart. If the duration is less than 24h, it is recorded 1 day; 24h-48h is 2 days, 48h-72h is 3 days, and so on. We have added an explanation of the exhaust time (Lines: 235-237).

15. Comments. The English must be improved. I suggest to ask for professional help.

Author response: We do think this is good suggestion. We have found a professional editing agency to modify the abstract, introduction and discussion parts of the article, and submit the receipt as an attachment. The methods and results of this article have been modified by high-level English users.

Minor comments:

1. Comments. Page 5, “The study was registered with ClinicalTrials.gov (ChiCTR 1800019479).”. It should read: “The study was registered on Chinese Clinical Trial Registry (ChiCTR 1800019479).”.

Author response: We do think this is good suggestion, which will make our paper more clarity. We have modified it.

2. Comments. Please expand TPVB the first time you used it.

Author response: We do think this is good suggestion, we have corrected it at Page 2 and Liens 9 .

3. Comments. Methods. The “Power analysis and sample size calculation” paragraph should be moved below, before statistical analysis paragraph.

Author response: We do think this is good suggestion, we have corrected it.

4. Comments. “dizzy” should probably reads “dizziness”

Author response: We do think this is good suggestion, we agree that the expression of dizziness is more professional, and we have corrected it (Table 2).

5. Comments. CONSORT diagram. You should report reason of discontinuation of intervention.

Author response: We do think this is good suggestion, we have corrected it.

We should like to express our appreciation to you for suggesting how improve our paper.

6. Review Comments to the Author

Dear Reviewer:

Thank you for your letter and for the comments concerning our manuscript entitled “Lower sedation index and raised blood glucose with opioid-free anesthesia compared to opioid anesthesia for lung cancer patients undergoing video-assisted thoracoscopic surgery: a prospective, randomized, controlled study” (PONE-D-20-27554). Those comments are all valuable and very helpful for revising and improving our paper, as well as the important guiding significance to our research. We have studied the comments carefully and have made corrections which we hope meet with approval. Revised portion were marked in yellow in the paper. The main corrections in the paper and the responses to the reviewer’s comments are as flowing:

Responses to the reviewer’s comments:

Reviewer #6: Thanks for the revisions to the paper. Many improvements were made, but there are still some points to clarify.

1. Comments: Pg 6. Line 100 (track changed version) I feel the statistical sample size calculation text in the manuscript does not adequately match with the words in the response to reviewers.

Author response: Thank you for pointing this out. This description is indeed inappropriate, and we have made changes (Lines 201-202, marked with yellow).

2. Comments:There is now more of a mismatch between 88 pts and 100 pts. Please explain.

Author response: We do think this is a good question. With a level of significance 0.05, a power 0.9, true difference 6 and SD 10, each group should include 44 patients. Considering that the postoperative follow-up was in the ward before discharge, the dropout rate was determined to be 10%-20%. The statistical instructor of this experiment considered that a sample size of 100 is acceptable.

3. Comments:The 10 patients who refused to participate - did they do so before or after randomisation? If so, say so, and specify the usual time frame between registration/ agreeing to participate and randomisation. This process needs further clarity.

Author response: We do think this is a good question. From November 2018 to March 2019, we included 222 patients who underwent thoracoscopic lobectomy. Among them, 35 were due to age > 65 years, 10 were due to BMI > 30%, and 5 had history of alcohol, 62 people had cardiopulmonary insufficiency, and 10 people refused to participate in the experiment. Eventually 100 participants entered the experiment. We have modified the relevant content, see Fig 1 and Lines 216. 

Therefore, 110 patients refer to the composite inclusion criteria, and there are no exclusion factors. Patients who refused to participate in the experiment were not included in the experiment. Eligible patients who agree to participate in the experiment will receive a sequentially numbered, opaque, sealed envelopes (Lines:111-112). 

Once again, Special thanks to you for your good comments and suggestions.

---

## [Decision Letter · Decision Letter 3]

27 May 2021

PONE-D-20-27554R3

Opioid-free anesthesia compared to opioid anesthesia for lung cancer patients undergoing video-assisted thoracoscopic surgery: a randomized controlled study

PLOS ONE

Dear Dr. Zhao,

Thank you for submitting your manuscript to PLOS ONE. After careful consideration, we feel that it has merit but does not fully meet PLOS ONE’s publication criteria as it currently stands. Therefore, we invite you to submit a revised version of the manuscript that addresses the points raised during the review process.

There are still several minor improvements to be performed (see below).  The language quality must be improved. We advise that you seek assistance from a colleague or have a professional editing service.

We look forward to receiving your revised manuscript.

Kind regards,

Alessandro Putzu, M.D.

Academic Editor

PLOS ONE

Journal Requirements:

Additional Editor Comments:

Thank you for your further work on this manuscript which now makes a fine contribution to consideration of this important topic. There are still several minor improvements to be performed.

1. You should insert authors contributions (see PLOS One guidelines)

2. I suggest to use a structured abstract (not exceeding 300 words)

3. Abstract. You should cite the EEG monitoring system in the abstract.

4. Abstract. The TPVB abbreviation should be removed from the abstract.

5. Abstract, line 20-24. It should read: “This study showed that our OFA regimen achieved equally effective intraoperative pain threshold index compared to OA in video-assisted thoracoscopic surgery. Depth of sedation was significantly lower and blood glucose levels were higher with OFA.”

6. Introduction, line 39. It should read: “…respiratory depression [3], postoperative nausea and vomiting (PONV) [4] as well as postoperative delirium…”

7. Methods, trial design, line 85-88. This paragraph should be removed: “Because midazolam is associated with postoperative delirium, midazolam is no longer used before anesthesia in patients undergoing thoracic surgery in our hospital. The eligible patient was informed that they are not received midazolam and signed a written informed consent form after obtaining the consent of the participant.”

8. Methods, allocation and blinding, line 97. You should not report results in the methods. It should read: “Patients were randomized in two groups (group OA and group OFA) using computer-generated random numbers (Excel® version 16)”

9. Methods, Anesthetic management protocol, page 8, line 140-141. It should read: “Sevoflurane was adjusted to maintain anesthesia. The use of midazolam was avoided. The patients were maintained on mechanical ventilation to keep the ….”

10. Methods, outcomes, page 10, line 183. Exhaust time should be defined

11. Methods, outcomes, page 10, line 185-190. The English of the paragraph should be improved.

12. Results, page 14, line 283-284. This sentence should be improved “No patient died, was transferred to ICU and respiratory dysfunction, and no patient was intubated again after the tracheal intubation was removed”. If judged correct, it should read: “No patient was transferred to the ICU, had respiratory insufficiency, need reintubation, or died during the follow-up”.

13. Discussion, first paragraph, line 290-295. It should read: “This prospective randomized study showed that our OFA regimen could provide equally adequate intraoperative analgesia-nociception balance in non-obese, 18-65 years old, ASA I-II patients compared to opioid based anesthesia guided by PTI monitoring in patients undergoing VATS. However, in group OFA, the intraoperative depth of sedation was deeper, and the blood glucose levels were higher, than those levels in patients with OA.”

14. Discussion, line 312-313. Ij judged correct, it should read: “…sevoflurane and TPVB were used in our study to replace opioids[26].”

15. Discussion, page 16, line 332. It should read: “…postoperatively improved pain relief (especially pain on coughing) up to 48 h….”

16. Discussion, page 17, line 358. It should read: “Our results suggested that, in patients under non-opioid anesthesia…”

17. Discussion, page 19, limitations paragraph, line 391. It should read: “Fourth, our study was a single-institution study, small sample size, and some protocol amendments were performed. More in details, no data on PI were reported since PI can be only used in conscious patients and the serum potassium concentration was not recorded, because some patients were given potassium supplementation, which potentially results in a selection bias. Fifth, we cannot perform an intention-to-treat analysis including the 3 patients excluded after randomization since we did not follow up these patients. Further studies will be needed to assess the long-term benefits of the avoidance of opioid use in lung cancer patients, as well as to assess the benefits of increasing survival.”

18. Conclusions, line 402-405. It should read: “OFA with dexmedetomidine and sevoflurane plus TPVB appears to be feasible for the management of intraoperative nociception in ASA I-II, 18-65 years old, non-obese patients undergoing VATS for lung cancer. However, a deeper sedation level was produced in OFA than in OA for the achievement of an optimal balance of analgesia-nociception.”

19. Manuscript. The language quality must be improved. We advise that you seek assistance from a colleague or have a professional editing service.

Reviewers' comments:

Reviewer's Responses to Questions

**Comments to the Author**

1. If the authors have adequately addressed your comments raised in a previous round of review and you feel that this manuscript is now acceptable for publication, you may indicate that here to bypass the “Comments to the Author” section, enter your conflict of interest statement in the “Confidential to Editor” section, and submit your "Accept" recommendation.

Reviewer #6: All comments have been addressed

7. PLOS authors have the option to publish the peer review history of their article (what does this mean?). If published, this will include your full peer review and any attached files.

Reviewer #6: No

---

## [Author Response · Author response to Decision Letter 3]

18 Jun 2021

1. Comments. You should insert authors contributions (see PLOS One guidelines)

Author response: Thank you for your suggestion. As suggested by reviewer, we have inserted author contributions in title page and submitted online.

2. Comments. I suggest to use a structured abstract (not exceeding 300 words)

Author response: We are grateful for the suggestion. To be more clearly and in accordance with the comment, we have converted the abstract into a structured abstract, see abstract with yellow labels.

3. Comments. Abstract. You should cite the EEG monitoring system in the abstract.

Author response: We agree with your comment. We have added the EEG monitoring system in the abstract, see Lines 14-15 with yellow labels.

4. Comments. Abstract. The TPVB abbreviation should be removed from the abstract.

Author response: Thank you for pointing this out. We have removed the TPVB abbreviation from the abstract, see Lines 11 with yellow labels.

5. Comments. Abstract, line 20-24. It should read: “This study showed that our OFA regimen achieved equally effective intraoperative pain threshold index compared to OA in video-assisted thoracoscopic surgery. Depth of sedation was significantly lower and blood glucose levels were higher with OFA.”

Author response: We do think these are excellent suggestions which makes the manuscript more rigorous and concise (Lines: 27-31, marked with yellow). 

6. Comments. Introduction, line 39. It should read: “…respiratory depression [3], postoperative nausea and vomiting (PONV) [4] as well as postoperative delirium…”

Author response: Thank you for pointing this out. We have added ‘postoperative’ in lines 47 and marked with yellow.

7. Comments. Methods, trial design, line 85-88. This paragraph should be removed: “Because midazolam is associated with postoperative delirium, midazolam is no longer used before anesthesia in patients undergoing thoracic surgery in our hospital. The eligible patient was informed that they are not received midazolam and signed a written informed consent form after obtaining the consent of the participant.”

Author response: We deeply appreciate the reviewer’s suggestion. We have deleted the paragraph (Lines:95-98, yellow marked), and added the sentence “The eligible patient signed a written informed consent form after obtaining the consent of the participant.” (Lines:106-107, yellow marked).

8. Comments. Methods, allocation and blinding, line 97. You should not report results in the methods. It should read: “Patients were randomized in two groups (group OA and group OFA) using computer-generated random numbers (Excel® version 16)”

Author response: We do think this is good suggestion. We have corrected it, see line 110.

9. Comments. Methods, Anesthetic management protocol, page 8, line 140-141. It should read: “Sevoflurane was adjusted to maintain anesthesia. The use of midazolam was avoided. The patients were maintained on mechanical ventilation to keep the ….”

Author response: We do think this is excellent suggestion. We have made changes in relevant places in the manuscript marked with yellow, line 156-157.

10. Comments. Methods, outcomes, page 10, line 183. Exhaust time should be defined

Author response: Thank you for pointing this out. We have added the definition of exhaust time (Lines: 191, marked in yellow).

11. Comments. Methods, outcomes, page 10, line 185-190. The English of the paragraph should be improved.

Author response: We apologize for the language problems in the manuscript. Language presentation was improved with assistance from a native English speaker with appropriate research background (Lines: 193-203, marked with yellow).

12. Comments. Results, page 14, line 283-284. This sentence should be improved “No patient died, was transferred to ICU and respiratory dysfunction, and no patient was intubated again after the tracheal intubation was removed”. If judged correct, it should read: “No patient was transferred to the ICU, had respiratory insufficiency, need reintubation, or died during the follow-up”.

Author response: Our deepest gratitude goes to you for your careful work and thoughtful suggestions that have helped improve this paper substantially (Lines: 303-307, marked with yellow).

13. Comments. Discussion, first paragraph, line 290-295. It should read: “This prospective randomized study showed that our OFA regimen could provide equally adequate intraoperative analgesia-nociception balance in non-obese, 18-65 years old, ASA I-II patients compared to opioid based anesthesia guided by PTI monitoring in patients undergoing VATS. However, in group OFA, the intraoperative depth of sedation was deeper, and the blood glucose levels were higher, than those levels in patients with OA.”

Author response: Thank you for your precious comments and advice. The comment is all valuable and extremely helpful for revising and improving our paper. We have revised the manuscript accordingly, see line 313-318and yellow-marked.

14. Comments. Discussion, line 312-313. Ij judged correct, it should read: “…sevoflurane and TPVB were used in our study to replace opioids[26].”

Author response: Thank you for pointing this out. We have re-written this sentence (Lines: 340, yellow-marked).

15. Comments. Discussion, page 16, line 332. It should read: “…postoperatively improved pain relief (especially pain on coughing) up to 48 h….”

Author response: Thank you for pointing this out. We have rewritten this sentence (Lines: 359-361, marked with yellow).

16. Comments. Discussion, page 17, line 358. It should read: “Our results suggested that, in patients under non-opioid anesthesia…”

Author response: We do think this is good suggestion, which will make our paper more clarity. We have modified it (Lines: 389, marked with yellow).

17. Comments. Discussion, page 19, limitations paragraph, line 391. It should read: “Fourth, our study was a single-institution study, small sample size, and some protocol amendments were performed. More in details, no data on PI were reported since PI can be only used in conscious patients and the serum potassium concentration was not recorded, because some patients were given potassium supplementation, which potentially results in a selection bias. Fifth, we cannot perform an intention-to-treat analysis including the 3 patients excluded after randomization since we did not follow up these patients. Further studies will be needed to assess the long-term benefits of the avoidance of opioid use in lung cancer patients, as well as to assess the benefits of increasing survival.”

Author response: Thank you very much for revising this part. You have given me a more rigorous understanding of English wording and sentence making. According to your comment, we have rewritten this part, see line 425-437 and marked with yellow.

18. Comments. Conclusions, line 402-405. It should read: “OFA with dexmedetomidine and sevoflurane plus TPVB appears to be feasible for the management of intraoperative nociception in ASA I-II, 18-65 years old, non-obese patients undergoing VATS for lung cancer. However, a deeper sedation level was produced in OFA than in OA for the achievement of an optimal balance of analgesia-nociception.”

Author response: Thank you for underlining this deficiency. This section was revised and modified according to the suggestion (Lines:440-442, marked with yellow).

19. Comments. Manuscript. The language quality must be improved. We advise that you seek assistance from a colleague or have a professional editing service.

Author response: Thank you for your careful review. We are deeply sorry for the inconvenience they caused in your reading. The manuscript has been thoroughly revised and rewritten by a native English speaker, so we hope it can meet the journal’s standard. The modified details have been marked in yellow.

20. Comments. Please review your reference list to ensure that it is complete and correct. If you have cited papers that have been retracted, please include the rationale for doing so in the manuscript text, or remove these references and replace them with relevant current references. Any changes to the reference list should be mentioned in the rebuttal letter that accompanies your revised manuscript. If you need to cite a retracted article, indicate the article’s retracted status in the References list and also include a citation and full reference for the retraction notice.

Author response: Thank you for your reminder, we have re-checked the references, and the changes are marked in yellow.

Note: The authors of this article, Nuoya Chen and Bingsha Zhao, have moved to Hengshui People’s Hospital and Tianjin Chest Hospital due to their job changes. We have revised their current addresses (see Title page),

 Thank you for your careful review. We really appreciate your efforts in reviewing our manuscript. We wish good health to you, your family, and community. Your careful review has helped to make our study clearer and more comprehensive.

---

## [Editor Report · Decision Letter 4]

22 Jun 2021

PONE-D-20-27554R4

Opioid-free anesthesia compared to opioid anesthesia for lung cancer patients undergoing video-assisted thoracoscopic surgery: a randomized controlled study

PLOS ONE

Dear Dr. Zhao,

Thank you for submitting your manuscript to PLOS ONE. After careful consideration, we feel that it has merit but does not fully meet PLOS ONE’s publication criteria as it currently stands. Therefore, we invite you to submit a revised version of the manuscript that addresses the points raised during the review process.

The text needs some further corrections. Clarity and English should be improved. I suggest to ask for professional help; the language editing certificate is old (Date: 16-Feb-2021),

You find several corrections and comments in the attached file. Please open it with Adobe Acrobat Reader.

We look forward to receiving your revised manuscript.

Kind regards,

Alessandro Putzu, M.D.

Academic Editor

PLOS ONE
---

## [Author Response · Author response to Decision Letter 4]

17 Jul 2021

Major comments:

1. Comments. Please add a reference. A good quality review on the topic (analgesia monitoring) would be of the great interest for the reader (Lines 69-71).

Author response: Thank you for your suggestion. As suggested by reviewer, we have inserted a review [17] in line 71 on analgesic index monitoring (Thomas Ledowski. Objective monitoring of nociception: a review of current commercial solutions. British Journal of Anaesthesia, 2019;123 (2): e312ee321).

2. Comments. This is unclear (Lines 177). Clarity should be improved. "First exhaust" should be replaced.

Author response: We are grateful for the suggestion. To be more clearly and in accordance with the comment, we have improved the exhaust time which is the time from returning to the ward to the patient complaining of the first breaking wind (Lines: 179-180).

3. Comments. The reason is different that reasin in Figure 1 caption. Please modify (Lines 211-212).

Author response: Sorry for our negligence. We have modified the reasons for exclusion from the analysis in Figure 1and marked with yellow.

4. Comments. It is unclear (Lines: 288-292). English and clarity should be improved.

Author response: Thank you for your careful review. The manuscript has been thoroughly revised and rewritten by Elsevier Language Editing, we hope it can meet the journal’s standard. The modified details have been marked in yellow (Lines:290-295).

5. Comments. This is a repetition of the first paragraph (Lines: 293-295)?

Author response: We do think these are excellent suggestions which makes the manuscript more rigorous and concise. We have improved this paragraph (Lines: 296-297, marked with yellow). 

6. Comments. This long paragraph should be shortened. Clarity should be improved (Lines: 354-378).

Author response: We do think these are excellent suggestions which makes the manuscript more rigorous and concise. We have abbreviated this part in lines 360-386 and marked with yellow.

7. Comments. The strict inclusion criteria and study's limitations limit may limit the external validity of the study (Lines: 404).

Author response: We deeply appreciate the reviewer’s suggestion. We have rewritten the sentence as We pay more attention to the changes in blood glucose levels of diabetic patients with OFA (Lines:413-414, yellow marked).

---

## [Editor Report · Decision Letter 5]

20 Jul 2021

PONE-D-20-27554R5

Opioid-free anesthesia compared to opioid anesthesia for lung cancer patients undergoing video-assisted thoracoscopic surgery: a randomized controlled study

PLOS ONE

Dear Dr. Zhao,

Thank you for submitting your manuscript to PLOS ONE. After careful consideration, we feel that it has merit but does not fully meet PLOS ONE’s publication criteria as it currently stands. Therefore, we invite you to submit a revised version of the manuscript that addresses the points raised during the review process.

We look forward to receiving your revised manuscript.

Kind regards,

Alessandro Putzu, M.D.

Academic Editor

PLOS ONE

Journal Requirements:

Additional Editor Comments:

Thank you for the improvements and the great work on the study. There are still some minor corrections to perform.

Here you find my suggestions:

1- Abstract, page 2, line 18-19. It should read: “One hundred patients were randomized: 3 patients were excluded due to discontinued intervention and 97 included in the final analysis”.

2- Abstract, page 3, line 23. It should read: “… the blood glucose levels in group OFA increased by 20% compared”

3- Abstract, conclusions. There is an error and information on limitations should be reported. It should read: “This study suggested that an OFA regimen achieved equally effective intraoperative pain threshold index compared to OA in video-assisted thoracoscopic surgery. Depth of sedation and blood glucose levels were higher with OFA. Study's limitations and strict inclusion criteria may limit the external validity of the study, suggesting the need of further randomized trials on the topic.”

4- Page 5, line 71. It should read: “A variety of analgesia monitors have been developed to determine the depth of pain during surgery [17].”

5- Page 6, line 92-99. It should read: “We included patients with ASA physical status I or II, aged 18-65 years who were scheduled for elective thoracoscopic radical resection of lung cancer. Exclusion criteria were: pregnancy; breastfeeding; allergy to any experimental drug or its excipients; β-blockers therapy and HR <50 bpm; body mass index (BMI) more than 30 kg m−2; central nervous system diseases (such as epilepsy, cerebral infarction, or cerebral hemorrhage history); and history of chronic pain, alcohol, or drug abuse. The eligible patients signed a written informed consent form after obtaining the consent of the participant.”

5- Figure 1. The name of each group is missing in the figure. Please report OFA et OA labels.

6-Figure 1. Reasons for exclusions are reported twice (in the figure and in the caption). Please remove reasons for exclusion from figure caption. It should read: “Fig. 1 – Study flow diagram. OFA, opioid-free anesthesia; OA, opioid anesthesia”

7- Page 11, line 213-216. It should read: “Three patients were excluded due to discontinued intervention: 1 case turned to open surgery, 1 peripheral venous catheter slid out during surgery, 1 patient needed re-exploration for postoperative intra-thoracic bleeding on postoperative day 1. Outcomes of these patients were not recorded in the postoperative period. The remaining 97 patients were included in the final analysis (Fig 1).”

8- Exhaust time definition is still unclear. “The time from returning to the ward to the patient complaining of the first breaking wind” is totally unclear. Clarity should be improved.

9- Page 15, line 303-304. It should read: “…of OFA for patients under general anesthesia.”

10- Page 20, line 396-397. It should read: “Fourth, our research is a study a single-institution study, with a small sample size”.

11- Page 20, conclusions. It should read: “OFA with dexmedetomidine and sevoflurane plus TPVB seems to be feasible for the management of intraoperative nociception in ASA I-II, 18-65-years old, non-obese patients undergoing VATS for lung cancer. However, a deeper sedation level was produced in OFA than in OA for the achievement of an optimal balance of analgesia-nociception. A significant elevation in blood glucose levels in OFA group was also found. Study's limitations and strict inclusion criteria may limit the external validity of the study, suggesting the need of further randomized trials on the topic.”

---

## [Author Response · Author response to Decision Letter 5]

22 Jul 2021

1. Comments. Abstract, page 2, line 18-19. It should read: “One hundred patients were randomized: 3 patients were excluded due to discontinued intervention and 97 included in the final analysis”.

Author response: Thank you for your suggestion. As suggested by reviewer, we have rewritten this sentence, line 17-19 and marked with yellow.

2. Comments. Abstract, page 3, line 23. It should read: “… the blood glucose levels in group OFA increased by 20% compared”

Author response: We are grateful for the suggestion. To be more clearly and in accordance with the comment, we have changed ascend to increased (Lines: 24, marked with yellow).

3. Comments. Abstract, conclusions. There is an error and information on limitations should be reported. It should read: “This study suggested that an OFA regimen achieved equally effective intraoperative pain threshold index compared to OA in video-assisted thoracoscopic surgery. Depth of sedation and blood glucose levels were higher with OFA. Study's limitations and strict inclusion criteria may limit the external validity of the study, suggesting the need of further randomized trials on the topic.”

Author response: We do think these are excellent suggestions which makes the manuscript more rigorous and concise. We have added “Study's limitations and strict inclusion criteria may limit the external validity of the study, suggesting the need of further randomized trials on the topic” (Lines 30-32 and marked with yellow).

4. Comments. Page 5, line 71. It should read: “A variety of analgesia monitors have been developed to determine the depth of pain during surgery [17].”

Author response: Thank you for your suggestion. The modified details have been marked in yellow (Lines:67).

5. Comments. Page 6, line 92-99. It should read: “We included patients with ASA physical status I or II, aged 18-65 years who were scheduled for elective thoracoscopic radical resection of lung cancer. Exclusion criteria were: pregnancy; breastfeeding; allergy to any experimental drug or its excipients; β-blockers therapy and HR <50 bpm; body mass index (BMI) more than 30 kg m−2; central nervous system diseases (such as epilepsy, cerebral infarction, or cerebral hemorrhage history); and history of chronic pain, alcohol, or drug abuse. The eligible patients signed a written informed consent form after obtaining the consent of the participant.”

Author response: We do think these are excellent suggestions which makes the manuscript more rigorous and concise. We have improved this paragraph (Lines: 89-97, marked with yellow). 

6. Comments. Figure 1. The name of each group is missing in the figure. Please report OFA et OA labels.

Author response: We do think these are excellent suggestions. We have added label of OFA and label of OA (Lines 222-223 and marked with yellow).

7. Comments. Figure 1. Reasons for exclusions are reported twice (in the figure and in the caption). Please remove reasons for exclusion from figure caption. It should read: “Fig. 1 – Study flow diagram. OFA, opioid-free anesthesia; OA, opioid anesthesia”

Author response: We deeply appreciate the reviewer’s suggestion. We have rewritten the sentence (Lines:219-223, yellow marked).

8. Comments: Page 11, line 213-216. It should read: “Three patients were excluded due to discontinued intervention: 1 case turned to open surgery, 1 peripheral venous catheter slid out during surgery, 1 patient needed re-exploration for postoperative intra-thoracic bleeding on postoperative day 1. Outcomes of these patients were not recorded in the postoperative period. The remaining 97 patients were included in the final analysis (Fig 1).”

Author response: We deeply appreciate the reviewer’s suggestion. We have rewritten the sentence (Lines:212-216, yellow marked).

9. Comments: Exhaust time definition is still unclear. “The time from returning to the ward to the patient complaining of the first breaking wind” is totally unclear. Clarity should be improved.

Author response: We are grateful for the suggestion. Since the patients in this study were sent back to the ward about 10min-20min after the tracheal tube was removed, the exhaust time was recorded from returning to the ward to the first breaking wind.

To be more clearly and in accordance with the comment, we have changed the exhaust time from the end of the operation to the first breaking wind (Lines: 179, 232 and marked with yellow).

10. Comments: Page 15, line 303-304. It should read: “…of OFA for patients under general anesthesia.”

Author response: We deeply appreciate the reviewer’s suggestion. We have corrected the sentence (Lines:299, yellow marked).

11. Comments: Page 20, line 396-397. It should read: “Fourth, our research is a study a single-institution study, with a small sample size”.

Author response: We deeply appreciate the reviewer’s suggestion. We have corrected the sentence (Lines:383, yellow marked).

12. Comments: 11- Page 20, conclusions. It should read: “OFA with dexmedetomidine and sevoflurane plus TPVB seems to be feasible for the management of intraoperative nociception in ASA I-II, 18-65-years old, non-obese patients undergoing VATS for lung cancer. However, a deeper sedation level was produced in OFA than in OA for the achievement of an optimal balance of analgesia-nociception. A significant elevation in blood glucose levels in OFA group was also found. Study's limitations and strict inclusion criteria may limit the external validity of the study, suggesting the need of further randomized trials on the topic.” 

Author response: We do think these are excellent suggestions which makes the manuscript more rigorous and concise. We have improved this paragraph (Lines: 396-404, marked with yellow).

---

## [Editor Report · Decision Letter 6]

27 Jul 2021

PONE-D-20-27554R6

Opioid-free anesthesia compared to opioid anesthesia for lung cancer patients undergoing video-assisted thoracoscopic surgery: a randomized controlled study

PLOS ONE

Dear Dr. Zhao,

Thank you for submitting your manuscript to PLOS ONE. After careful consideration, we feel that it has merit but does not fully meet PLOS ONE’s publication criteria as it currently stands. Therefore, we invite you to submit a revised version of the manuscript that addresses the points raised during the review process.

We look forward to receiving your revised manuscript.

Kind regards,

Alessandro Putzu, M.D.

Academic Editor

PLOS ONE

Journal Requirements:

Additional Editor Comments (if provided):

1. Exhaust time is not an appropriate english medical term; you should change it. What does it mean? Do you mean " Time to passage of flatus and stool"? "Time to passage of flatus"? " Time to passage of stool"? Please modify

2. The details in Figure 1 are still missing (see PDF). You should include the denomination of each group.

3. Some other modifications are reported in the PDF attached.
---

## [Author Response · Author response to Decision Letter 6]

1 Aug 2021

1. Comments. Exhaust time is not an appropriate english medical term; you should change it. What does it mean? Do you mean " Time to passage of flatus and stool"? "Time to passage of flatus"? " Time to passage of stool"? Please modify.

Author response: We do think these are excellent suggestions which makes the manuscript more rigorous and concise. Anal exhaust is one of the indications of postoperative intestinal function recovery. The anal exhaust time was defined from the end of operation to the first spontaneous breaking wind event (Lines: 176-177, 228-229, Table 2, marked with yellow).

2. Comments. The details in Figure 1 are still missing (see PDF). You should include the denomination of each group.

Author response: Sorry for the omission in Figure 1, we have added denomination of each group (Figure 1).

3. Comments. Some other modifications are reported in the PDF attached.

Author response: We deeply appreciate the reviewer’s suggestion. We have made modifications according to the annotations in the PDF and marked with yellow.

---

## [Editor Report · Decision Letter 7]

16 Aug 2021

PONE-D-20-27554R7

Opioid-free anesthesia compared to opioid anesthesia for lung cancer patients undergoing video-assisted thoracoscopic surgery: a randomized controlled study

PLOS ONE

Dear Dr. Zhao,

Thank you for submitting your manuscript to PLOS ONE. After careful consideration, we feel that it has merit but does not fully meet PLOS ONE’s publication criteria as it currently stands. Therefore, we invite you to submit a revised version of the manuscript that addresses the points raised during the review process.

We look forward to receiving your revised manuscript.

Kind regards,

Alessandro Putzu, M.D.

Academic Editor

PLOS ONE

Journal Requirements:

Additional Editor Comments (if provided):

1. “ Anal exhaust time” is not a current appropriate English medical term. Please use "Time to passage of flatus"

2. I received a comment from Staff Editor. There are some deviations between the protocol, registry entry, and the manuscript. This it should be further explained by the authors. You can add the detailed list of protocol changes in the supplementary material. Most important changes should be reported in the methods. If you have any queries or need any help, please contact plosone@plos.org.

3. You should review your reference list to ensure that it is complete and correct, as reported above. No retracted manuscript should be cited.

---

## [Author Response · Author response to Decision Letter 7]

22 Aug 2021

1. Comments. “Anal exhaust time” is not a current appropriate English medical term. Please use "Time to passage of flatus".

Author response: We do think this is excellent suggestion which makes the manuscript more rigorous and concise. Thank you very much for pointing out our unprofessional use of words. The time to passage of flatus you provided is a standard expression. (Lines: 176-177, 229-230, Table 2, marked with yellow).

2. Comments. I received a comment from Staff Editor. There are some deviations between the protocol, registry entry, and the manuscript. This it should be further explained by the authors. You can add the detailed list of protocol changes in the supplementary material. Most important changes should be reported in the methods. If you have any queries or need any help, please contact plosone@plos.org.

Author response: Sorry for the inconvenience caused by the protocol changes. We summarize the changes in this research protocol in a table and submit it as a supplement (S File 9). The intraoperative PI and serum potassium are important changes that are not recorded. We explain the reasons and write them in the method and discussion section of the manuscript (Lines: 180-186, 385-387 and marked with yellow).

3. Comments. You should review your reference list to ensure that it is complete and correct, as reported above. No retracted manuscript should be cited.

Author response: We deeply appreciate the reviewer’s suggestion. We have made modifications according to Plos One's reference requirements and marked with yellow.

---

## [Editor Report · Decision Letter 8]

31 Aug 2021

Opioid-free anesthesia compared to opioid anesthesia for lung cancer patients undergoing video-assisted thoracoscopic surgery: a randomized controlled study

PONE-D-20-27554R8

Dear Dr. Zhao,

We’re pleased to inform you that your manuscript has been judged scientifically suitable for publication and will be formally accepted for publication once it meets all outstanding technical requirements.

Thank you for your further work on this manuscript which now makes a fine contribution to consideration of this important topic.

Kind regards,

Alessandro Putzu, M.D.

Academic Editor

PLOS ONE

---

## [Editor Report · Acceptance letter]

15 Sep 2021

PONE-D-20-27554R8 

Opioid-free anesthesia compared to opioid anesthesia for lung cancer patients undergoing video-assisted thoracoscopic surgery: a randomized controlled study 

Dear Dr. Zhao:

I'm pleased to inform you that your manuscript has been deemed suitable for publication in PLOS ONE. Congratulations! Your manuscript is now with our production department. 

Kind regards, 

on behalf of

Dr. Alessandro Putzu 

Academic Editor

PLOS ONE